# Attractive serial dependence overcomes repulsive neuronal adaptation

**Timothy C. Sheehan**[1]*, **John T. Serences**[1,2,3]

**1** Neurosciences Graduate Program, University of California San Diego, La Jolla, California, United States of America, **2** Department of Psychology, University of California San Diego, La Jolla, California, United States of America, **3** Kavli Institute for Brain and Mind, University of California San Diego, La Jolla, California, United States of America

* tsheehan@health.ucsd.edu

**Data Availability Statement:** All data and code required to complete analysis are available on an OSF repository at https://osf.io/e5xw8/?view_only=e7c1da85aa684cc8830aec8d74afdcb4.

**Funding:** National Institute of Mental Health (https://www.nimh.nih.gov/) Training Grant in

## Abstract

Sensory responses and behavior are strongly shaped by stimulus history. For example, perceptual reports are sometimes biased toward previously viewed stimuli (serial dependence). While behavioral studies have pointed to both perceptual and postperceptual origins of this phenomenon, neural data that could elucidate where these biases emerge is limited. We recorded functional magnetic resonance imaging (fMRI) responses while human participants (male and female) performed a delayed orientation discrimination task. While behavioral reports were attracted to the previous stimulus, response patterns in visual cortex were repelled. We reconciled these opposing neural and behavioral biases using a model where both sensory encoding and readout are shaped by stimulus history. First, neural adaptation reduces redundancy at encoding and leads to the repulsive biases that we observed in visual cortex. Second, our modeling work suggest that serial dependence is induced by readout mechanisms that account for adaptation in visual cortex. According to this account, the visual system can simultaneously improve efficiency via adaptation while still optimizing behavior based on the temporal structure of natural stimuli.

## Introduction

Natural stimuli are known to have strong statistical dependencies across both space and time, such as a prevalence of vertical and horizontal (cardinal) orientations and a higher probability of small orientation changes in given spatial region over short time intervals [1–4]. These regularities can be leveraged to improve the efficiency and accuracy of visual information processing. For example, regularities can yield attenuated neural responses to frequently occurring stimuli in early visual cortex (adaptation), reducing metabolic cost and redundancy in neural codes [5–9]. At readout, regularities might support the formation of Bayesian priors that can be used to bias decision-making in favor of higher probability stimuli [10–12]. While the effects of stimulus history on sensory coding and behavior have been studied extensively, it is unclear how changes in sensory coding shape behavior.

Adaptation increases coding efficiency by modulating sensory tuning properties as a function of the recent past. For example, reducing the gain of neurons tuned to a recently seen

Cognitive Neuroscience (T32- MH020002) to TCS National Eye Institute (https://www.nei.nih.gov/) R01-EY025872 and National Institutes of Mental Health (https://www.nimh.nih.gov/) R01-MH087214 to JTS The funders had no role in study design, data collection and analysis, decision to publish, or preparation of the manuscript.

**Competing interests:** The authors have declared that no competing interests exist.

**Abbreviations:** AUC, area under the curve; CCW, counterclockwise; CW, clockwise; DoG, Derivative of Gaussian; EEG, electroencephalography; EPI, echo-planar imaging; FEF, frontal eye field; fMRI, functional magnetic resonance imaging; FWHM, full width at half maximum; GLM, generalized linear model; HRF, hemodynamic response function; IEM, inverted encoding model; ISI, interstimulus interval; ITI, intertrial interval; MEG, magnetoencephalography; MLE, maximum likelihood estimation; PCA, principal component analysis; ROI, region of interest; RSS, residual sum of squared errors.

adapting stimulus reduces the temporal autocorrelation of activity when similar stimuli are presented sequentially, improving the overall efficiency of sensory codes [7,13–16]. Importantly, adapted representations early in the processing stream (e.g. the Lateral Geniculate Nucleus, LGN) are inherited by later visual areas, meaning the changes in coding properties could, in turn, shape decision-making [8,17,18]. Although adaptation increases coding efficiency, it comes at a cost to perceptual fidelity as adaptation can lead to repulsion away from the adapting stimulus for features such as orientation and motion direction [19–21]. For example, after continuously viewing and adapting to motion in one direction, stationary objects will appear to be moving in the opposite direction (i.e., current perceptual representations are repelled away from recent percepts). However, this potentially deleterious aftereffect is accompanied by better discriminability around the adapting stimulus, which may be more important than absolute fidelity from a fitness perspective [16,22–24].

In contrast to the repulsive perceptual biases typically associated with neural adaptation, perceptual reports are sometimes attracted to recently presented items—a phenomenon termed "serial dependence." Studies utilizing low contrast oriented stimuli suggest that serial dependence can be perceptual in nature as it operates before a peripheral tilt illusion, impacts the perception of simultaneously presented items, biases perceptual reports even when no probe is presented, and does not require a working memory delay [25–29]. This perceptual account could arise from activity changes in early visual cortex, consistent with a functional magnetic resonance imaging (fMRI) study that measured early sensory biases that match "attractive" behavioral reports [30]. This neural finding, however, is challenging to interpret as consecutive trials were always the same or orthogonal orientations, which, by definition, cannot distinguish attractive from repulsive biases. Related studies decoding past stimuli from electroencephalography (EEG) activity do not measure how current stimulus representations are biased, precluding a connection to behavioral biases [31–33].

Counter to studies reporting a perceptual locus of serial dependence that utilized brief or low contrast stimuli, other behavioral studies utilizing high-contrast spatial stimuli have found that serial dependence does not emerge immediately but instead emerges only, and increases with, a working memory maintenance period [34–36]. This observation suggests that serial dependence could be implemented by a later readout or memory maintenance circuit [34,37–39]. There is evidence that such a readout mechanism is Bayesian, as the influence of the "prior" (the previous stimulus) is larger when sensory representations are less precise due to either external or internal noise [4,40]. Thus, the existing behavioral evidence suggests that serial dependence can operate both on perceptual and working memory representations [26,34,41]. It is open question how and where past trial information interacts with incoming sensory and memory representations.

To determine what role visual cortex plays in driving serial dependence, we applied multivariate fMRI decoding techniques to data collected while participants performed a delayed orientation discrimination task (Fig 1A). We replicated classic serial dependence findings where behavioral reports were attracted to the orientation of the previous stimulus. However, this attractive behavioral bias was not accompanied by attractive biases in visual cortex, as predicted by early sensory models of serial dependence. Rather, we observed repulsive biases in early visual cortex that were consistent with adaptation. We then examined several possible read-out mechanisms and found that only decoding schemes that account for adaptation can reconcile the neural and behavioral biases found in our data. More generally, these results explain a mechanism where the visual system can reduce energy usage without sacrificing precision by optimizing sensory coding and behavioral readout relative to the temporal structure of natural environments.

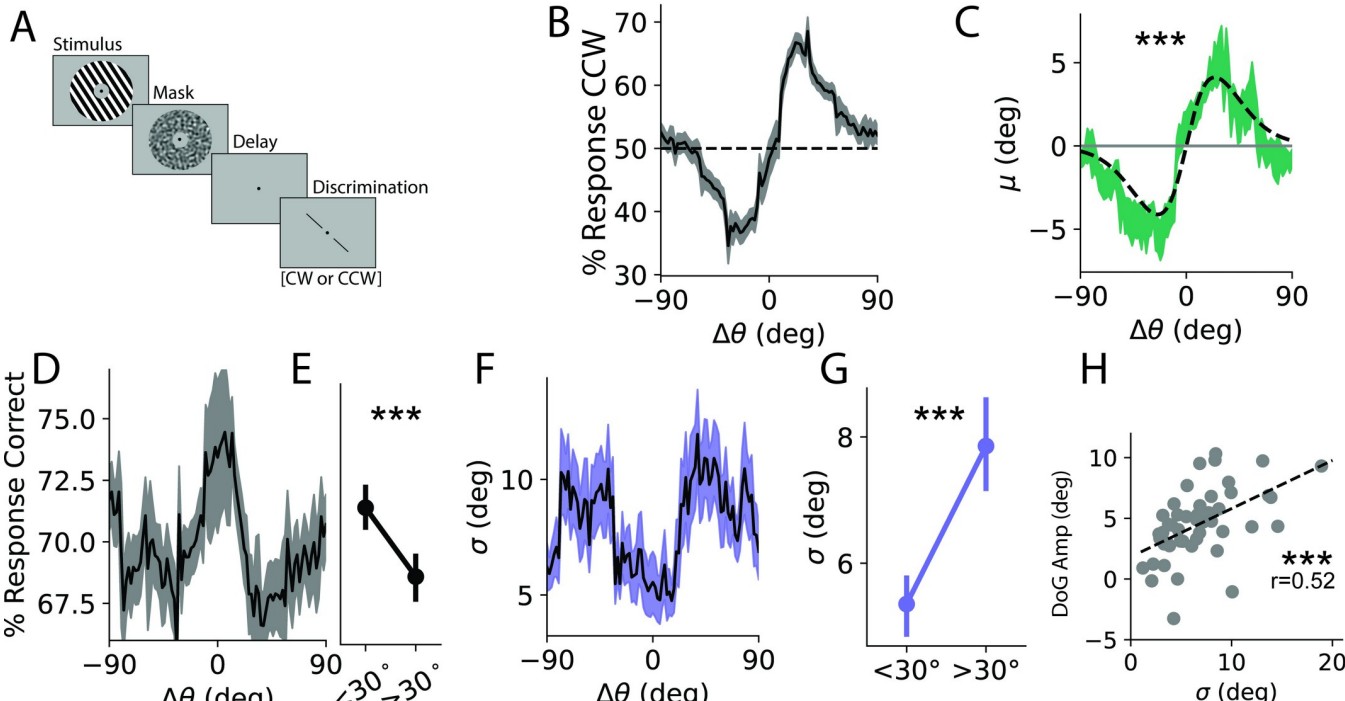

**Fig 1. Caption behavior.** **(A)** Task schematic. An orientated stimulus is followed by a probe bar that is rotated <15˚ from the stimulus. Participants judged whether the bar was CW or CCW relative to the stimulus in a binary discrimination task. **(B)** Response bias: % of responses that were CCW as a function of Δθ = θn − 1 − θn (± SEM across participants). **(C)** Behavioral bias, green: average model-estimated bias as a function of Δθ (± SEM across participants); gray: average DoG fit to raw participant responses sorted by Δθ (± 1SEM across participants). **(D)** Response accuracy as a function of Δθ. **(E)** Responses are significantly more accurate for |Δθ|<30˚. **(F)** Behavioral σ as a function of Δθ. **(G)** Behavioral variance is significantly less for |Δθ|<30˚. Note that in computing variance, we "flip" the sign of errors following CCW inducing trials to avoid conflating bias with variance (see Methods). **(H)** Bias is positively correlated with variance across participants. ***, $p < 0.001$. Data and code supporting this figure found here: https://osf.io/e5xw8/?view_only= e7c1da85aa684cc8830aec8d74afdcb4. CCW, counterclockwise; CW, clockwise.

## Results

### Behavior

To probe the behavioral effects of serial dependence, we designed a delayed discrimination task where participants judged whether a bar was tilted clockwise (CW) or counterclockwise (CCW) relative to the orientation of a remembered grating (Fig 1A). We first report the results from a behavior-only study ($n$ = 47) followed by an analysis of neural activity for a cohort completing the same task in the fMRI scanner ($n$ = 6). Task difficulty was adjusted for each participant by changing the magnitude of the probe offset (δθ) from the remembered grating and was titrated to achieve a mean accuracy of approximately 70% (accuracy 69.8 ± 0.82%, δθ: 4.61 ± 0.27˚; all reported values mean ± 1 SEM unless otherwise noted). Fixing participants at this intermediate accuracy level helped to avoid floor/ceiling effects and improved our sensitivity to detect perceptual biases while keeping participants motivated.

To quantify the pattern of behavioral responses, we modeled the data as the product of a noisy encoding process described by a Gaussian distribution centered on the presented orientation with standard deviation σ and bias μ. Optimal values for σ and μ were found by maximizing the likelihood of responses for probes of varying rotational offsets from the remembered stimulus, thus converting pooled binary responses into variance and bias measured in degrees (see Response bias; S1 Fig). This allowed us to measure precision for individual participants and also allowed us to measure how responses were biased as a function of the

orientation difference between the remembered gratings on consecutive trials $\Delta\theta = \theta_{n-1} - \theta_n$, an assay of serial dependence.

Responses were robustly biased toward the previous stimulus (Fig 1C, green curve), which we quantified by fitting a Derivative of Gaussian (DoG) function to the raw response data for each participant (gray curve; amplitude: 4.53˚ ± 0.42˚, $t(46) = 7.8$, $p = 5.9^*10^{-10}$, 1-sample $t$ test; full width at half maximum (FWHM): 42.9˚ ± 1.8˚; see Serial dependence). The magnitude and shape of serial dependence are consistent with previous reports [25,42]. This bias is not an artifact of our parameterization as the same pattern is observable in the raw proportion of CCW responses (Fig 1B). Note that as participants are reporting the orientation of the probe relative to the grating stimulus, a greater proportion of reports that the probe was CCW corresponds to a CW shift in the perception of the grating.

We next examined how response precision (σ) varied as a function of Δθ and found that responses were more precise around small trial-to-trial orientation changes (Fig 1F), again consistent with previous reports [43]. We quantified this difference in precision by splitting trials into "close" and "far" bins (greater than or less than 30˚ separation) and confirmed that responses following "close" stimuli were more precise ($t(46) = -3.72$, $p = 0.0003$, paired 1-tailed $t$ test, Fig 1G; see Response precision). Note that the choice of 30˚ was arbitrary, but all threshold values between 20˚ and 40˚ yielded significant ($p < 0.05$) results. As with bias, this variance result was not an artifact of our parameterization as raw accuracy showed a similar pattern such that responses were more accurate following close stimuli (t(46) = 3.66, $p = 0.0003$; Fig 1D and 1E). We additionally confirmed that our finding of reduced bias around small changes in orientation is not driven by a higher proportion of "cardinal" orientations (here defined as being ±22.5˚ of 0 or 90˚) as the proportion of cardinal orientations did not differ between close and far bins of Δθ (mean % cardinal close: 50.6 ± 0.5%, far: 50.2 ± 0.3%, t(46) = 0.9, $p = 0.39$, paired $t$ test).

Previous work has shown that serial dependence is greater when stimulus contrast is lower [28] and when internal representations of orientation are weaker due to stimulus independent fluctuations in encoding fidelity [4]. We tested a Bayesian interpretation of these findings by asking whether less precise individuals are more reliant on prior expectations and therefore more biased. Consistent with this account, we found a positive correlation between DoG amplitude and σ (Fig 1H, r(45) = 0.52, $p = 0.0001$, 1-tailed Pearson correlation). This relationship was not dependent on our response parameterization as we report found similar relationships between DoG amplitude and both accuracy (r = −0.41, Pearson correlation, $p < 0.005$) and average task difficulty δθ (r = 0.44, $p < 0.005$).

A subset of participants completed a version of the experiment with inhomogeneities in their stimulus sequences (such that consecutive orientations were more likely to be between ±22.5 and 67.5˚ from the previous stimulus). We repeated all of the above analyses excluding these participants and found all of our findings were qualitatively unchanged (S2 Fig).

## Stimulus history effects in visual cortex

To examine the influence of stimulus history on orientation-selective response patterns in early visual cortex, 6 participants completed between 748 and 884 trials (mean 838.7) of the task in the fMRI scanner over the course of four 2-hour sessions (average accuracy of 67.7% ± 0.4% with an average probe offset, δθ, of 3.65˚). As with the behavior-only cohort, behavioral reports in these participants showed strong attractive serial dependence (Fig 2A, green) that was significantly greater than 0 when parameterized with a DoG function (amplitude = 3.50˚ ± 0.27˚, $t(5) = 11.93$, $p = 0.00004$; FWHM = 35.9˚ ± 2.34˚, Fig 2A, black dotted line). This bias was not significantly modulated by intertrial interval, delay period, or an interaction between

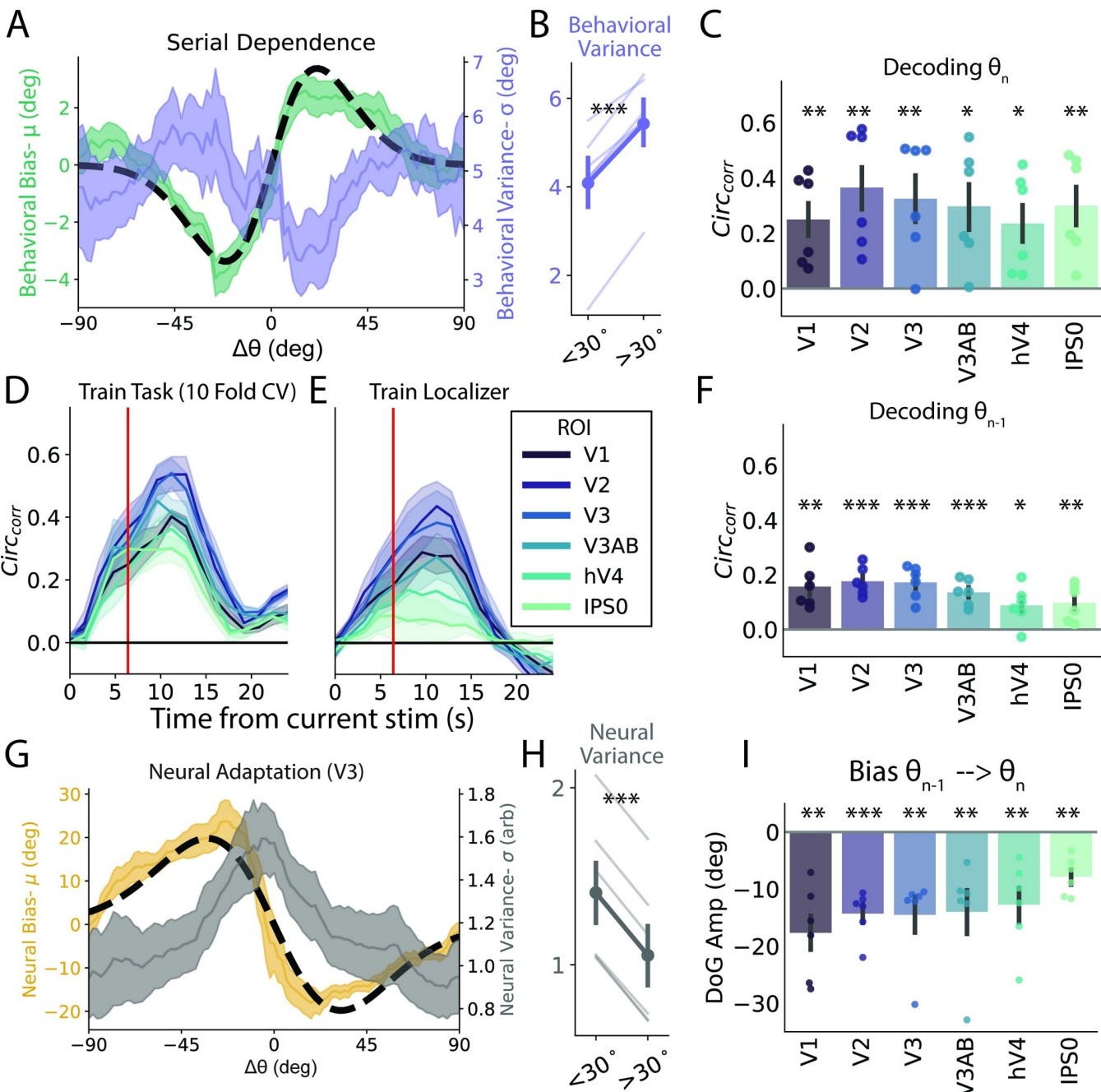

**Fig 2. Caption behavioral and neural bias. (A)** Left axis, behavioral serial dependence. Shaded green: average model-estimated bias as a function of Δθ (±
SEM across participants); dotted black line: average DoG fit to raw participant responses sorted by Δθ. Right axis, variance. Purple shaded line: model-
estimated variance as a function of Δθ (± SEM across participants). **(B)** Behavioral σ is significantly less for |Δθ|<30˚. **(C)** Decoded orientation was significantly
greater than chance when indexed with circular correlation for all ROIs examined. Error bars indicate ±SEM across participants. Dots show data from
individual participants. **(D)** Decoding performance across time for a subset of ROIs. Vertical red line indicates time point used in most analysis. **(E)** Decoding
performance across time for a decoder trained on a separate sensory localization task. **(F)** Performance of task decoder trained and tested on identity of
previous stimulus across all ROIs. **(G)** Left axis, decoding bias. Shaded yellow line: decoded bias ($\mu_{circ}$ of decoding errors) sorted by Δθ (± SEM across
participants); dotted black line: average DoG fit to raw decoding errors sorted by Δθ. Right axis, decoded $\sigma_{circ}$. Shaded gray line: average decoding variance
($\sigma_{circ}$) as a function of Δθ (± SEM across participants). Note that $\sigma_{circ}$ can range from [0, inf] and has no units. **(H)** Decoded variance is significantly greater for
|Δθ|<30˚. **(I)** Decoded errors are significantly repulsive when parameterized with a DoG in all ROIs. *, $p < 0.05$; **, $p < 0.01$; ***, $p < 0.001$. Data and code
supporting this figure found here: https://osf.io/e5xw8/?view_only=e7c1da85aa684cc8830aec8d74afdcb4. DoG, Derivative of Gaussian; ROI, region of interest.

the 2 factors (all $p$-values $> 0.5$, mixed linear model grouping by participant). Similar to the behavioral cohort, we found that variance was generally lower around small values of Δθ. We quantified variance in the same manner as the behavioral cohort (flipping responses to match biases and down-sampling the larger group) and found that responses were more precise following close ($< 30˚$) relative to far stimuli ($> 30˚$, $t(5) = −9.96$, $p = 0.00009$, 1-tailed paired $t$ test, Fig 2B). This pattern was significant ($p < 0.05$) for thresholds between $20˚$ and $40˚$. A subset of these participants completed some sessions where consecutive stimuli were not strictly independent as they were more likely to be between ±22.5 and 67.5˚ from the previous stimulus (see Methods, Behavioral discrimination task, 4 out of 6 participants had between 357 and 408 trials that were nonindependent accounting for between 40% and 50% of their trials and 32% of all trials completed). However, we replicated all of our main analysis excluding these sessions and found that our conclusion remained unchanged with the exception that our finding of reduced variance trended in the same direction but no longer reached significance (S3 Fig).

To characterize activity in early visual areas, independent retinotopic mapping runs were completed by each participant to identify regions of interest (ROIs) consisting of V1, V2, V3, V3AB, hV4, and intraparietal sulcus area IPS0. In addition, a separate localizer task was used to subselect the voxels that were most selective for the spatial position and orientation of the stimuli used in our task (see Voxel selection).

To examine how visual representations are affected by stimulus history, we trained a decoder on the orientation of the sample stimulus on each trial based on BOLD activation patterns in each ROI. We used the vector mean of the output of an inverted encoding model (IEM) as a single trial measure of orientation using a leave-one-run-out cross-validation across sets of 68 consecutive trials (4 blocks of 17 trials) that had orientations pseudo randomly distributed across all 180˚ of orientation space (see Orientation decoding for details). We first quantified single-trial decoding performance using circular correlation ($r_{circ}$) between the decoder-estimated orientations and the actual presented orientations and found that all ROIs had significant orientation information (Fig 2C). Our ability to decode extended for the duration of the trial, peaking around 12 seconds after stimulus presentation (Fig 2D). This memory signal seems to be largely in a "sensory code" as a decoder trained on a separate localizer task where participants viewed stimuli without holding them in memory achieved similar performance over a similar timescale (see fMRI localizer task; Fig 2E). Thus, visual ROIs showed robust orientation information that could be decoded across the duration of the trial. For all analyses not shown across time, we used the average of 4 TRs (repetition time, spanning 4.8 to 8.0 seconds) following stimulus presentation to minimize the influence of the probe stimulus (which came up ≥6 seconds into the trial and thus should have a negligible influence on activity in the 4.8 to 8.0 seconds window after accounting for hemodynamic delay; see Fig 5A).

We are interested in the how the identity of the previous stimulus influences representations of the current stimulus, akin to previous EEG studies that have demonstrated the ability to decode the previous stimulus during the current trial [32]. We performed a similar analysis by training and testing our task decoder on the identity of the previous stimulus using the same time points as the current trial decoder. This decoder was able to achieve above chance decoding in all ROIs examined indicating trial history information is present in the activity patterns (Fig 2F). As a control analysis, we attempted but were unable to decode the identity of the next stimulus using the same procedure (S3F Fig). The performance of the memory decoder for the previous stimulus peaked around 6 seconds after stimulus presentation but remained above chance throughout the delay period (S4A Fig). Notably, we were generally unable to decode the identity of the previous stimulus using our decoder trained on a localizer task suggesting representations of past trial stimuli are not in a "sensory code" (S4B Fig).

The high SNR (signal to noise ratio) of the BOLD decoder additionally allowed us to examine residual errors on individual trials. When measuring the bias (circular mean, $\mu_{circ}$; see Neural bias) of these decoding errors as a function of stimulus history ($\Delta\theta$), we observed a strong repulsive bias reflecting neural adaptation (V3, Fig 2G, yellow). This bias was significant when quantified with a DoG (amplitude = $-14.5° \pm 2.9°$, $t(5) = -3.56$, $p = 0.0029$; FWHM = $52.2° \pm 2.94°$, Fig 2G, black dotted line), and all ROIs had a significantly negative amplitude ($p < 0.01$, Fig 2I). Critically, this bias was present across all TRs for both the task and localizer decoders and was visible in the bias curve computed for each individual participant (S4 Fig). In addition to the model-based analysis of responses in visual cortex, we also performed a model-free assessment of the dimensionality of activation patterns conditioned on the prior stimulus. Consistent with our main analysis, responses following close stimuli have a higher dimensionality than responses following far stimuli. This suggests that changes due to neural adaptation should assist pattern separation regardless of stimulus identity (see Dimensionality analysis; S5 Fig).

We also examined how the precision of neural representations changed as a function of stimulus history. In sharp contrast to behavior, $\sigma_{circ}$ exhibited a monotonic trend such that neural decoding was least precise when the previous stimulus was similar (Fig 2G, gray curve; see Neural variance). We quantified this difference in sensory uncertainty in a similar manner to the behavioral data and found that variance in the sensory representations was significantly greater following a similar stimulus ($<30°$, $t(5) = 72.4$, $p = 4.8*10^{-9}$, paired 1-tailed $t$ test, V3, Fig 2H). This pattern was significant ($p < 0.05$) in all ROIs except IPS0 (S6A Fig). The results did not change qualitatively when we utilized vector length as a proxy for decoding precision derived directly from our channel estimates (S6C and S6D Fig) or when we used other thresholds between $20°$ and $40°$. The repulsion of sensory representations and the corresponding reduction in decoding precision around the previous orientation is consistent with neural adaptation where recently active units are attenuated, thus leading to lower SNR responses in visual cortex.

## Accounting for the time course of the hemodynamic response function

We considered whether the repulsive adaptation we observed in visual cortex could be explained by residual undershoot of the hemodynamic response function (HRF) from the previous stimulus. To address this concern, we directly modeled the evoked response in each voxel to the stimulus and probe using a deconvolution approach and used a parameterization of the resulting filter (double gamma function) to model the stimulus evoked response on each trial (see Kernel-based decoding). Notably, the stimulus response has an undershoot that extends up to 25 seconds following stimulus presentation (see Fig 3A for an example voxel and parameterization). Estimating responses using this filter on individual trials and using the resulting weights to train a decoder removes the linear contribution of previous stimulus/probe presentations [44,45]. Any bias in the resulting decoder should thus be due to changes in BOLD activity driven by neuronal activity rather than a hemodynamic artifact. We repeated all analyses after correcting for the shape of the HRF, and while the resulting decoder was less precise than one trained on the time course data (eg. V3 $r_{circ} = 0.19 \pm 0.07$ versus $0.32 \pm 0.08$ with time course decoder), it was still significantly predictive across all visual ROIs ($ps < 0.05$) except IPS0. Despite the noisier decoding, we still observed a significant repulsive bias in all visual ROIs that matched the pattern found when decoding the raw BOLD time course (Fig 3B).

To further understand whether the time course of our task could lead to artifacts, we also simulated responses to our task using tuned voxels that were modeled after the task sequence and estimated HRFs observed in our experiment (see supplementary modeling section,

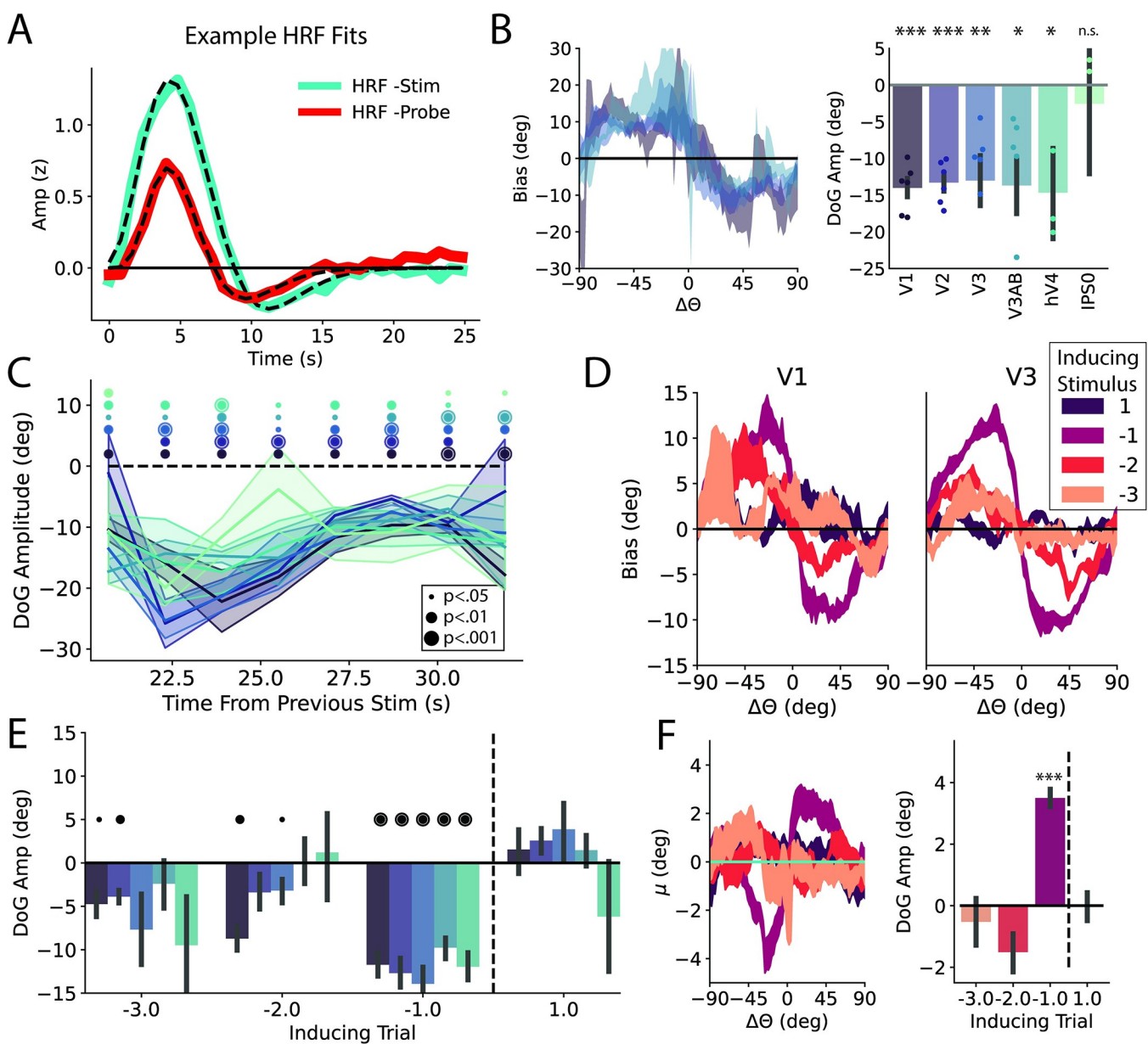

**Fig 3. Influence of BOLD-specific biases on repulsive bias.** (**A**) Average V1 HRF through deconvolution for stimulus and probe. Average best fit double gamma function overlaid in dotted lines. (**B**) (Left) Bias curves from decoder trained on response patterns from deconvolved double-gamma functions (± SEM across participants). Here excluding hV4 and IPS0 for clarity. (Right) Bias quantified with a DoG function across ROIs. (**C**) Bias across time including only trials with an ISI of at least 17.5 seconds. x-Axis reflects minimum time from previous stimulus. Repulsion significant in all ROIs at 32 seconds. (**D**) Bias as a function of various relative orientations for V1 and V3 (± SEM across participants). (**E**) Bias across early visual ROIs for N-1, N-2, and N-3. Color scheme same as C. N+1 control analysis to ensure effects not driven by some unknown structure in stimulus sequence. (**F**) Behavioral bias for various relative orientations. N-1 data same as data presented in Fig 2. *, $p < 0.05$, **, $p < 0.01$, ***, $p < 0.001$. Data and code supporting this figure found here: https://osf.io/e5xw8/?view_only=e7c1da85aa684cc8830aec8d74afdcb4. DoG, Derivative of Gaussian; HRF, hemodynamic response function; ROI, region of interest.

S7 Fig). These simulations show that repulsive biases like the ones we observed with both our time course and deconvolution-based decoders are only possible when the underlying tuning of voxels is adapted by past stimuli/responses.

We additionally examined the time-course of the bias. Significant repulsive biases were observable through the duration of the trial, in all early visual ROIs (S4 Fig). As the undershoot

portion of the HRF extended to approximately 25 seconds, we examined the bias relative to the time of the presentation of the previous stimulus. We included only trials with an inter-stimulus interval (ISI) greater than the median of 17.5 seconds and plotted bias as a function of the minimum time from the previous stimulus (Fig 3C). Notably, bias was still significantly repulsive for 30 seconds following the previous stimulus presentation in all early visual ROIs, further shrinking the possibility that our biases are driven by the slow time course of the HRF (Fig 3C, last time point). Finally, we examined how far back previous stimuli shape early visual representations. We examined the influence of not just the N-1 stimulus, but N-2 and N-3 stimuli as well, corresponding to median ISIs of 35.1 and 52.5 seconds, respectively (Fig 3D and 3E). As any influence of these more distant stimuli should be diminished relative to N-1, we maximized our sensitivity by taking the average decoded representation from 4 to 12 seconds. While the control N+1 stimulus showed no impact on decoded orientation as expected, we continued to see biases that are significantly repulsive through the N-3 stimulus in V1 and V2 (Fig 3E). These neural biases were surprisingly persistent and are in line with recent studies that have found adaptation signatures extending 22 seconds in mouse visual cortex spiking activity [9]. It is not clear why our effects persist even longer, but it is likely driven in part by the long ISIs, resulting in fewer intervening stimuli compared to the paradigm utilized in [9]. We separately extended our analysis of behavioral biases and found no significant effect of trials except for N-1, although biases were trending toward being repulsive for N-2 and N-3 reflecting the pattern reported in [46] (Fig 3F). Together, these analyses suggest that our observed biases are driven by adaptation in the underlying neural population and provide additional evidence that behavior is not directly linked to early visual representations.

## Encoder–decoder model

We observed an attractive bias and low variability around the current stimulus feature in behavior, and a repulsive bias and high variability around the current feature in the fMRI decoding data. Thus, the patterns of bias and variability observed in the behavioral data are opposite to the patterns of bias and variability observed in visual cortex. To better understand these opposing effects, we reasoned that representations in early visual cortex do not directly drive behavior but instead are read out by later cortical regions that determine the correct response given the task [47–50]. In this construction, the decoded orientations from visual cortex represent only the beginning of a complex information processing stream that, in our task, culminates with the participant making a speeded button press response. Thus, we devised a 2-stage encoder–decoder model to describe observations in both early visual cortex and in behavior (see Modeling).

The encoding stage consists of cells with uniformly spaced von Mises tuning curves whose amplitude is adapted by the identity of the previous stimulus ($\theta_{n-1}$, Fig 4A). The decoding stage reads out this activity using 1 of 3 strategies (Fig 4B). The unaware decoder assumes no adaptation has taken place and results in stimulus likelihoods $p(m|\theta)$ that are repelled from the previous stimulus (Fig 4C, yellow, where m is the population activity at the encoding stage). This adaptation-naive decoder is a previously hypothesized mechanism for behavioral adaptation [51] and likely captures the process that gives rise to the repulsive bias we observe in visual cortex using a fMRI decoder that is agnostic to stimulus history (Fig 2G). Alternatively, the *aware* decoder (Fig 4C, green) has perfect knowledge of the current state of adaptation and can thus account for and "undo" biases introduced during encoding. Finally, the overaware decoder knows the identity of the previous stimulus but overestimates the amount of gain modulation that takes place, resulting in a net attraction to the previous stimulus (Fig 4C, red). We additionally built off of previous work showing stimuli are generally stable across time by

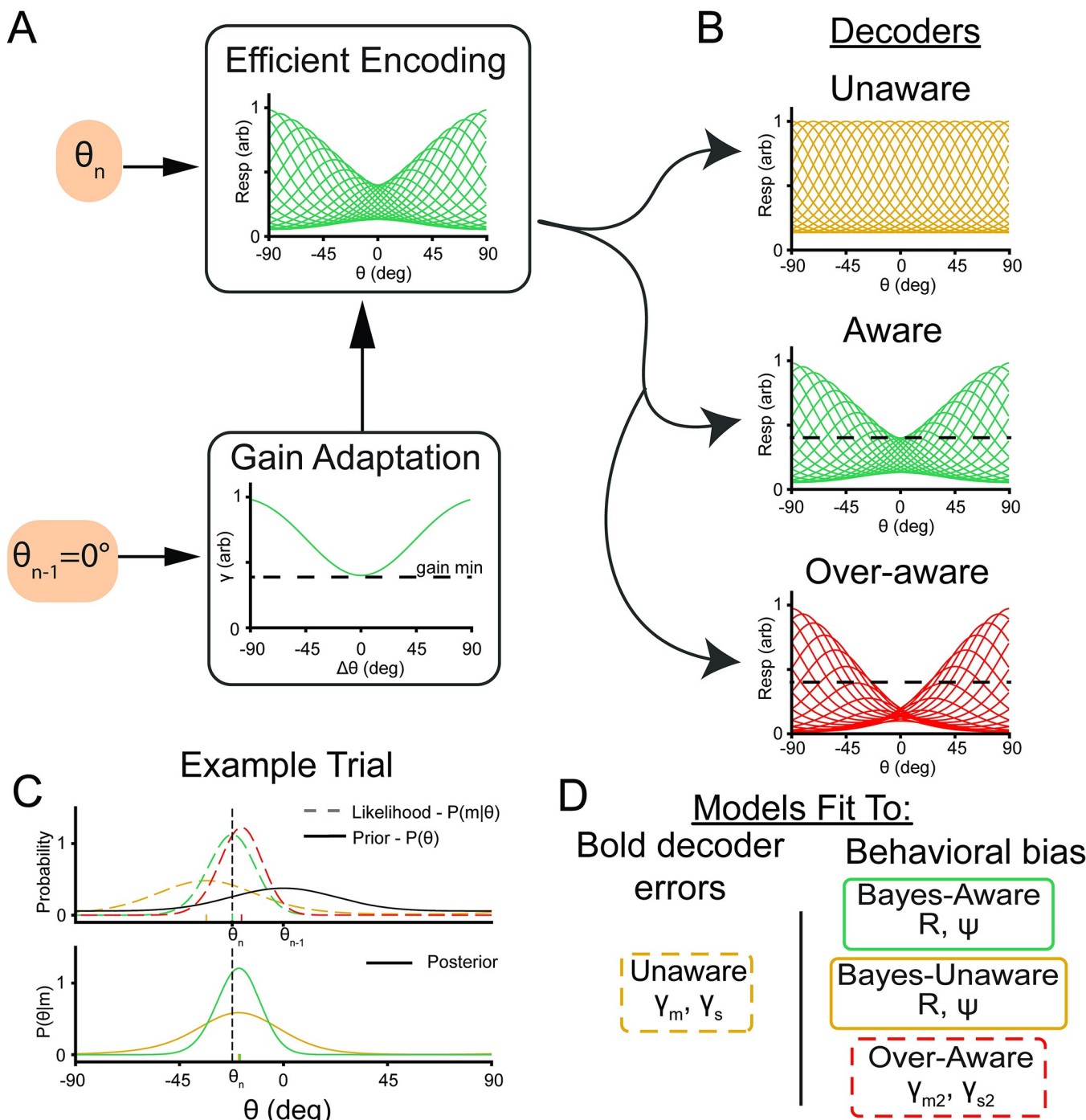

**Fig 4. Encoder–decoder model schematic. (A)** Encoding. Units with von Mises tuning curves encodes incoming stimuli. The gain of individual units undergoes adaptation such that their activity is reduced as a function of their distance from the previous stimulus. **(B)** Decoding. This activity is then read out using a scheme that assumes 1 of 3 adaptation profiles. The unaware decoder assumes no adaptation has taken place, the aware decoder assumes the true amount of adaptation while the overaware decoder overestimates the amount of adaptation (note center tuning curves dip lower than the minimum gain line from encoding). **(C)** Example stimulus decoding. Top: The resulting likelihood function for the unaware readout (dotted yellow line) has its representation for the current trial ($\theta_n = -30°$) biased away from the previous stimulus ($\theta_{n-1} = 0°$). The aware readout (dotted green line) is not biased, while the overaware readout is biased toward the previous stimulus. These likelihood functions can be multiplied by a prior of stimulus contiguity (solid black line) to get a Bayesian posterior (bottom) where Bayes-unaware and Bayes-aware representations are shifted toward the previous stimulus. Tick marks indicate maximum likelihood or decoded orientation. **(D)** Summary of models and free parameters being fit to both BOLD decoder errors and behavioral bias. Data and code supporting this figure found here: https://osf.io/e5xw8/?view_only=e7c1da85aa684cc8830aec8d74afdcb4.

implementing a prior of temporal contiguity [4]. In our implementation, a Bayesian prior centered on the previous stimulus (Fig 4C, black) is multiplied by the decoded likelihood to get a Bayesian posterior (Fig 4C, bottom). We applied this prior of temporal contiguity to both the aware decoder as well as the unaware decoder to test the importance of awareness at decoding. We did not apply a prior to the overaware model to balance the number of free parameters between the various decoders and to see if the overaware model could achieve attractive serial dependence without a Bayesian prior (Fig 4 and S1 Table).

For each participant, we fit the encoder–decoder model in 2 steps (Fig 4D). All model fitting was performed using the same cross-validation groups as our BOLD decoder and each stage had 2 free parameters that were fit using grid-search and gradient descent techniques. We first report results from the encoding stage of the model. The gain applied at encoding was adjusted to minimize the residual sum of squared errors (RSS) between the output of the unaware decoder and the residual errors of our BOLD decoder. The unaware readout of the adapted encoding process (Fig 5A, yellow) provided a good fit to the average decoding errors obtained with the BOLD decoder (Fig 5A, black outline, $\rho = 0.99$) and across individual participants (S8A Fig, ranges: $\rho = [0.84, 0.98]$). The unaware readout provided a better fit to the outputs of our neural decoder than the null alternative of the presented orientation ($t(5) = 3.41$, $p = 0.01$) because it captured a significant proportion of the variance in decoding errors as a function of $\Delta\theta$ ($t(5) = 7.5$, $p = 0.0007$). This analysis demonstrates that our adaptation model does a reasonable job of recovering our empirical decoding data (both of which use a decoder unaware of sensory history).

We next considered 3 readout schemes of this adapted population to maximize the likelihood of our behavioral responses (Fig 5B). The Bayes-aware decoder is consistent with previous Bayesian accounts of serial dependence [4], but additionally asserts that Bayesian inference occurs after encoding and that readout must account for adaptation. Alternatively, the Bayes-unaware decoder tests whether this awareness is necessary to achieve attractive serial dependence. Both aware models achieved biases that were significantly more likely than the unaware model ($t(5) = 6.53$, $p = 0.001$, Bayes-aware; $t(5) = 6.6$, $p = 0.001$, overaware, $t$ test on log-likelihood, Fig 5C) but were indistinguishable from each other ($p = 0.36$). Thus, both aware models were able to explain the response biases while the unaware model did a relatively poor job, suggesting that some awareness of the adapted state is necessary.

Finally, we examined the variance of our decoders to see if this mapped onto our empirically observed variance. As model coefficients were fit independent of observed variance, correspondence between model performance and BOLD/behavioral data would provide convergent support for the best model. While the models were trained using noiseless activity at encoding, we simulated responses using Poisson rates to induce response variability. We simulated 1,000 trials from each cross-validated fit and pooled the model outputs. We first confirmed that the variance of the unaware decoder was highest following small changes of $\Delta\theta$ (Fig 5A, gray; Fig 5G $t(5) = 3.93$, $p = 0.005$, paired 1-tailed $t$ test $<30°$ versus $>30°$) matching the output of our neural decoder (Fig 2G) and providing additional support for gain adaptation causing the observed repulsion in the fMRI data. Next, we compared the different decoders and found that, matching real behavioral responses, all 3 decoders were more precise following small values of $\Delta\theta$ (Fig 5G, Bayes-unaware, $t(5) = 2.25$, $p = 0.037$; Bayes-aware $t(5) = 1.90$, $p = 0.058$; and overaware $t(5) = 5.43$, $p = 0.001$). While the pattern of the Bayes-unaware variance matched behavior, its overall variance was much higher than our behavioral data such that it diverged from the behavioral data significantly more than either of the aware models (Fig 5E–5G; $ps < 0.005$, paired $t$ test comparing Jenson–Shannon divergence of error distributions). Together, the variance data provide additional evidence in favor of adaptation driving the repulsive biases that were observed in the BOLD data and awareness of the current

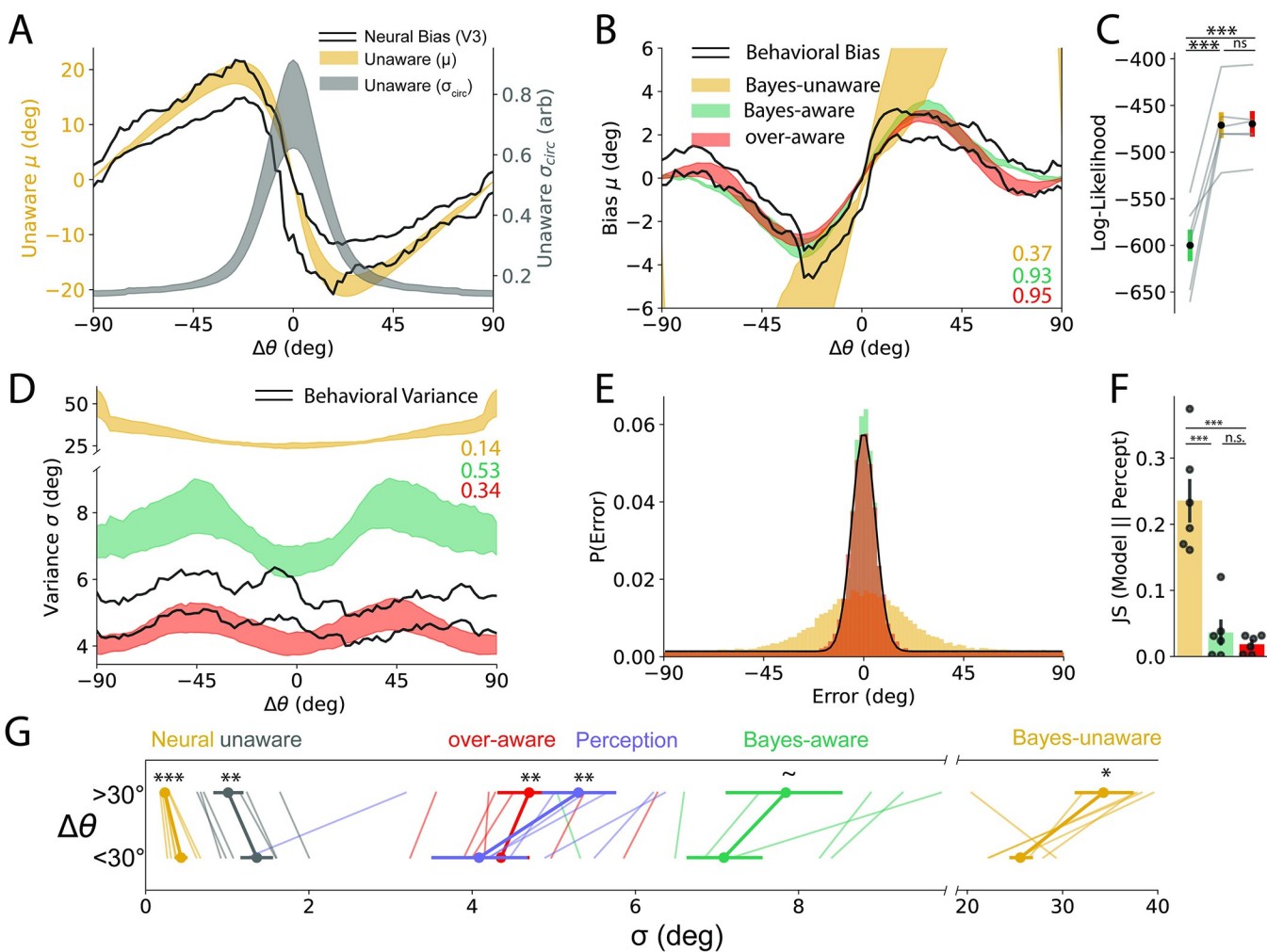

**Fig 5. Model performance bias. (A–C)** Neural/behavioral bias. **(D–G)** Neural/behavioral variance. (A) Unaware decoder (yellow) provides a good fit to neural bias (black outline). Decoded variance decreases monotonically with distance from previous stimulus. (± SEM across participants). (B) Perceptual bias (black outline) was well fit by the Bayes-aware and overaware models but not the Bayes-Unaware model (± SEM across participants). (C) Participant responses were significantly more likely under aware models. (D) Behavioral variance had a similar shape and magnitude to Bayes-aware and overaware model fits. Bayes-unaware model output was much less precise and had a different form. (E) Distribution of empirically predicted response errors (black line) and simulated model fits for an example participant. (F) The unaware model's error distribution had significantly higher Jenson–Shannon divergence from BOLD decoder than either aware model. (G) Visualization of all uncertainties split as a function of close and far stimuli. Note that the Bayes-unaware model had an average uncertainty that was on average 6x that of perception. *, $p < 0.05$; **, $p < 0.01$; ***, $p < 0.001$. Data and code supporting this figure found here: https://osf.io/e5xw8/?view_only=e7c1da85aa684cc8830aec8d74afdcb4.

state of adaptation being a requisite condition for the observed attractive serial dependence. More generally, this model has notable advantages that can lead to enhanced discrimination, reduced energy usage, and improved discrimination in naturalistic conditions over a static labeled line representation.

## Discussion

In this study, we sought to understand the neural underpinning of attractive serial dependence and how changes in tuning properties at encoding shape behavior. Based on previous behavioral and neural studies, we expected to observe attractive biases in line with observed behavior and decoding from early visual areas [30]. Instead, we found that representations were

significantly repelled from the previous stimulus starting in primary visual cortex and continuing through IPS0 (Fig 2I). This repulsion is consistent with bottom up adaptation beginning either at or before V1 and cascading up the visual hierarchy [8,9,18]. As repulsive biases are in the opposite direction as behavioral biases, we built a model to link these conflicting patterns. The critical new insight revealed by the model is that only readout schemes that account for adaptation can explain the attractive behavioral bias observed in our paradigm. More generally, our BOLD data argue against an early sensory origin of serial dependence for orientation and instead suggest that serial dependence is driven by postperceptual or mnemonic circuits [38,39]. However, because we used a paradigm that required working memory, our results may not generalize to other situations in which serial dependence is observed even in the absence of a memory delay [25,26,29,52]. Thus, future work is needed to better understand the role of sensory representations in paradigms with low contrast stimuli, that do not require a memory delay period, and that utilize other features besides orientation.

There have been many prior studies arguing for either a perceptual or postperceptual origin of serial dependence. Some behavioral studies have found that serial dependence emerges almost immediately after the offset of a stimulus, pointing to an early perceptual origin of the effect [25–27,40]. One study additionally demonstrated that attraction to the previous stimulus seems to occur before the "tilt illusion" driven by concurrently presented flanking stimuli [27]. If history biases indeed operate before spatial context, this could point to a distinct assimilative mechanism for serial dependence in early visual processing that may only emerge under low stimulus drive. As our experiment always utilizes a working memory delay, it is unclear if the bias toward past stimuli is driven by a change in their perception of the stimulus itself or instead somehow biases their comparison with the probe stimulus only after the working memory maintenance period.

Others have found that serial dependence is repulsive at very short delays and only becomes attractive when items are held for an extended time in working memory [34,35]. This apparent discrepancy was reconciled by [28], who showed that attractive biases disappear without a working memory delay, unless the stimuli are rendered at a very low contrast. This observation suggests that serial dependence may emerge immediately when high sensory uncertainty is induced by low contrast stimuli, and it may emerge later if high sensory uncertainty is induced by extended working memory delay periods. It is curious that unlike some spatial working memory studies [34–36], we did not find that behavioral biases increased with delay time. One possible explanation is that this phenomenon is actually unique to spatial working memory due to either a more consistent increase in sensory uncertainty of spatial location due to eye movements or a separate mechanism of memory maintenance that becomes more susceptible to proactive interference relative to orientation memories. Separately, as our stimuli were presented at the fovea (unlike spatial paradigms) they are encoded by a larger population and thus may be less susceptible to degradation across time.

Evidence for an early sensory origin of serial dependence comes from an fMRI study with low contrast stimuli and a short (500 ms) delay period which reported that both behavioral responses and V1 representations were more precise following a matching stimulus [30]. This departure from our own finding could be driven by the stimuli that were rendered to have a very high uncertainty. Past work studying adaptation in nonhuman primates found repulsive patterns following long (4 seconds and 40 seconds) but attractive patterns following short (0.4 seconds) stimulus presentations, suggesting that stimulus duration may have a large influence on how past stimuli shape future sensory processing [53]. That said, the stimuli used in the fMRI study of [30] were always 1 of 2 orthogonal orientations, which, given a circular feature space like orientation, precludes an assessment of attraction or repulsion. Furthermore, correct motor responses were directly yoked to the stimulus so any behavioral tendency to report

seeing the same stimulus on successive trials could be due to motor priming rather than stimulus based serial dependence (e.g., a "stay" bias). Related work has shown the ability to decode the previous stimulus from EEG activity patterns [31–33], but it is important to note that our study also showed robust decoding of the previous stimulus that did not also correspond with an attractive bias in the neural representation of the current stimulus (Figs 2F and 2G and S4). This is because the representations of current and past stimuli are not necessarily stored using the same code. Thus, while previous neural studies have argued that serial dependence emerges in visual cortex, no study has demonstrated an attraction toward the previous stimulus dependent on feature similarity consistent with behavioral biases. Further work examining neural biases using low contrast stimuli will shed further light on a potential role of coding changes in sensory cortex driving serial dependence.

In contrast to studies favoring an early sensory account—and more in line with the paradigm and findings reported in this manuscript—a single unit recording study in nonhuman primates used high-contrast stimuli and a longer working memory delay (1.4 to 5.6 seconds) [54]. Under these conditions, neural responses in the frontal eye fields (FEFs) were repelled from the previously remembered location even though saccades were attracted to the previously remembered location. Given the tight link between the FEF and attentional control [55–57], the authors speculated that the observed neural repulsion was due to residual attentional shifts carrying over from the previous trial. However, our observation of repulsive biases starting in V1 and persisting across later visual areas suggests that bottom-up adaptation may be a viable alternative explanation (which the authors also acknowledged). Further support for this account comes from a recent magnetoencephalography (MEG) study showing that representations were repelled from past stimuli both within the current trial and from the previous trial [58]. As in our study, this neural repulsion contrasts with attractive behavioral biases to the previous stimulus, suggesting that sensory representations do not directly shape behavior even in simple sensory paradigms [50]. Behavioral studies using similar high-contrast orientation stimuli to our own have also shown that responses are attracted to past decisions and repelled from past stimuli, further suggesting that these attractive biases do not emerge in early sensory areas [38,59,60]. Several modeling studies additionally suggest that serial biases are mediated by later readout circuits due to synaptic changes arising from persistent bump attractor dynamics as opposed to early sensory processing [37,39]. Thus, in line with our findings: behavioral, neuronal, and modeling studies utilizing high-contrast stimuli in working memory paradigms consistently point to attractive effects emerging in either memory or decision-making circuits and not early sensory areas.

In line with classic accounts, adaptation in visual cortex should lead to a reduction in energy usage during encoding [14]. However, the main advantage of adaptation may be to decorrelate inputs, thus enhancing the discriminability of incoming stimuli [14,61] and even acting as a form of short-term memory [62]. An optimal processing stream may emphasize differences at encoding and only favor stability once a stimulus has been selected by attention for more extensive postperceptual processing [38]. This motif of pattern separation followed by pattern completion would not be unique to adaptive visual processing. Similar mechanisms have been proposed as a critical component of long-term memory processing in the hippocampus and associative memory formation in the fly mushroom body [63]. Thus, the biases introduced by adaptation may be beneficial in part because they expand the dimensionality of the representational space as we found in our recordings (S5 Fig).

We did not explicitly define how awareness of adaptation is implemented, but it is clear that both attention to and conscious awareness of the previous stimulus are necessary for serial dependence to occur [25,64]. This is consistent with our model, and it suggests that some representation of information about stimulus history should be a minimum requirement for

an aware decoding scheme. The identity of the previous stimulus for spatial position and angle has been shown to be decodable from the spiking activity of single units in the FEF and posterior parietal cortex (PPC) as well as large-scale activity patterns in human EEG and MEG [31–33,54,58,65]. We additionally demonstrate that information about the previous trial is encoded in patterns of fMRI activity in human visual cortex (Fig 2F), but not in a sensory-like code (S4A and S4B Fig). These signals could potentially be represented concurrently with representations of the current stimulus in the same populations of sensory neurons but in orthogonal codes analogous to what has been found for sequentially encoded items in primate prefrontal cortex and human EEG [66,67]. An alternate account holds that representations of stimulus history are maintained outside of early visual areas, consistent with findings from mouse parietal and primate prefrontal cortex [39,65]. This anatomical segregation could disambiguate incoming sensory drive from representations of stimulus history. Critically, optogenetically suppressing nonsensory representations of stimulus history eliminated history effects, thus providing strong support for some form of an aware readout mechanism [65].

For the decoding stage of our model, we established that only readout schemes that are aware of adaptation could explain attractive serial dependence. The Bayes-aware model is an extension of previously proposed models that employ an explicit prior but that did not consider effects of adaptation at encoding [4]. In contrast, the overaware model is a novel account that can achieve similar performance without needing an explicit prior based on stimulus history. While model fit metrics did not readily distinguish one of these 2 models as superior, the overaware model may prove to be more flexible. For example, one of our fMRI participants showed significant repulsion from far stimuli, an observation also reported by others [35,42]. While the overaware model can fit this repulsive regime, the Bayes-aware model is incapable of generating repulsive patterns (compare models fits for participant #3; S8 Fig). This limitation of a purely Bayesian account of serial dependence is also observable in prior work (Fig 6B in [4]).

The overaware model proposed in our study may instead be a special condition of a decoder with "fixed awareness" that is based on temporal transition probabilities in natural scenes that are steeply peaked around 0 (no change) over short timescales [1,2,4]. Such a readout would correct for the most encountered levels of adaptation by accounting for the transition probabilities of stimuli while being "fixed," or inflexible, when stimuli violate these expectations. This decoder could account for additional phenomena not directly assessed in the present study such as the tilt aftereffect (TAE). The TAE and other forms of (repulsive) behavioral adaptation are often ascribed to an unaware decoder [7,51] but might instead reflect levels of adaptation that exceed the fixed level of adaptation expected by a "fixed-aware" decoder due to long presentations or high-contrast stimuli. This is supported by an apparent disconnect in the magnitude of repulsive biases between behavior and neural representations [5,19]. In contrast, the fixed awareness decoder would lead to attractive biases (serial dependence) when stimuli create less bottom-up drive than expected (e.g., through brief presentations or low contrast items). This "fixed-aware" decoder is consistent with previous findings of attractive biases disappearing or switching to repulsion when stimulus contrast or duration is increased [25,28]. This scheme could extend to spatial adaptation such as the tilt illusion where the joint probability of center and surround orientations being perfectly distinct would be vanishingly rare in natural scenes [68–70].

In this study, we extended previous descriptions of serial dependence by quantifying how both bias and variance are shaped by stimulus history. We report a robust pattern of perception being most precise following small changes in successive stimulus features (Figs 1F, 1G, 2A and 2B). This relationship violates a proposed perceptual "law" that bias is inversely proportional to the derivative of discrimination thresholds [71]. This account would assert that

our attractive bias should come with a less precise representation following small changes (or a repulsive bias to account for our enhanced precision). We argue that serial dependence is not violating this law, but rather believe this is further evidence for delay dependent serial dependence being a postsensory phenomenon. Neural representations exhibit repulsive biases, expanding the perceptual space and allowing greater discriminability (S5 Fig). When these representations are read out by an aware decoder, the bias is undone but the enhanced discriminability remains (Fig 5D and 5G).

## Methods

### Participants

Behavioral study: A total of 56 participants (male and female) were drawn from a participant pool of primarily undergraduate students at UC San Diego. All participants gave written consent to participate in the study in accordance with the UC San Diego IRB (approval number 180067) and were compensated either monetarily or with class credit. Of these 56 participants, 9 were removed from further analysis for completing less than 200 trials (2) or getting less than 60% of trials correct (7). We included the remaining 47 participants who completed on average 421 trials, range: [204, 988], in our lab over the course of 1 to 3 sessions.

fMRI study: A total of 6 participants (3 female, mean age 24.6 ± 0.92) participated in four 2-hour scanning sessions. Each participant completed between 748 and 884 trials (mean 838.7). For 2 participants, 1 session had to be repeated due to technical difficulties that arose during scanning.

### Behavioral discrimination task

Participants in the behavior-only study completed the task on a desktop computer in a sound attenuated room. Participants were seated with a chin rest to stabilize viewing 50 cm from a 39 by 29 cm CRT monitor (1600 × 1200 px) with a visual angle of 42.6˚ (screen width). Each trial consisted of a full-field oriented grating (1,000 ms), which had to be remembered across a delay period (3,500 ms) before a test. At test, the participant judged whether a line was slightly CW or CCW relative to the remembered orientation (max response time window: 3,000 ms, Fig 1A). The oriented grating consisted of a sine wave grating (spatial frequency 1.73 cycles/˚, 0.8 Michelson contrast) multiplied by a "donut" mask (outer diameter Ø = 24.3˚, inner Ø = 1.73˚). The stimulus was then convolved with a 2D Gaussian filter (1.16˚ kernel, SD = 0.58˚) to minimize edge artifacts [72]. Phase and orientation were randomized across trials, and the stimulus was phase-reversed every 250 ms. After the offset of the oriented grating, a mask of filtered noise was presented for 500 ms. The mask was generated by band passing white noise [low 0.22, high 0.87 cycles/˚], multiplying by the same donut mask, and convolving with a 2D Gaussian filter (0.27˚ kernel, SD = 0.11˚). The mask was phase reversed once after 250 ms. A black fixation point (diameter 0.578˚) was displayed throughout the extent of the block and turned white for 500 ms prior to stimulus onset on each trial. The probe was a white line (width 0.03˚, length 24.3˚) masked by the same donut. Participants indicated whether the probe line was CW or CCW from the remembered orientation by pressing 1 of 2 buttons ("Q," "P") with their left and right pointer fingers. The next trial started after a 1,000 ms intertrial interval (ITI). For some behavioral participants (n = 9), delay and ITI were varied between 0.5 and 7.5 seconds without notable effects on performance.

First, participants completed a training block to ensure that they understood the task. Next, they completed a block of trials where difficulty was adjusted by changing the probe offset (δθ) between the stimulus and probe to achieve 70% accuracy. This δθ was used in subsequent blocks and was adjusted on a per-block basis to keep performance at approximately 70%.

Participants completed an average of 5.76 ± 0.24 blocks [min = 3, max = 9]. Some participants completed the task with slight variations in the distribution and sequence of orientations presented. For completeness, we include those details here. Note, however, we additionally report a set of control analyses in which we repeat all of our main analyses excluding blocks with binned stimuli and find no relevant difference in behavior. For most participants, stimuli were pseudo-randomly distributed across the entire 180˚ space such that they were uniformly distributed across blocks of 64 trials (n = 25). However, some participants saw stimuli that were binned (with some jitter) every 22.5˚ to purposefully avoid cardinal and oblique orientations (11.25˚, 33.75˚, 56.25˚, etc.), and the trial sequence was ordered so that a near oblique orientation was always followed by a near cardinal orientation (n = 7). This was implemented to maximize our ability to observe serial dependencies in our binary response data as it is typically strongest around orientation changes of 20˚ and is more pronounced around oblique orientations [43]. The remaining participants completed both blocks with uniform and blocks with binned stimuli (n = 14). All participants were interviewed after the study and reported that stimuli were nonpredictable and that all orientations felt equally likely. For our main analysis, we include all trials from all participants, irrespective of whether they participated in uniform blocks, binned blocks, or both.

## fMRI discrimination task

In the scanner, participants completed the behavioral task outlined above with slight modifications. fMRI participants completed the task using a fiber-optic button box while viewing stimuli through a mirror projected onto a screen mounted inside of the bore. The screen was 24 by 18 cm and was viewed at a distance of 47 cm (width: 28.6˚ visual angle; $1024 \times 768$ px native resolution). The stimulus timing was the same except that the sample-to-probe delay period was either 5, 7, or 9 seconds, and the ITIs were uniformly spaced between 5 seconds and 9 seconds and shuffled pseudo-randomly on each run of 17 trials. The oriented gratings had a spatial frequency of 1.27 cycles/˚, outer Ø = 21.2˚, inner Ø = 2.37˚ and were smoothed by a Gaussian filter (0.79˚ kernel, sd = 0.79˚). The noise patch (SF low 0.16, high 0.63 cycles/˚) was also smoothed by a Gaussian filter (0.29˚ kernel, sd = 0.11˚). The probe stimulus was a white line (width = 0.03˚).

fMRI participants completed 44 to 52 blocks of 17 trials spread across four 2-hour scanning sessions for a total of 748 to 884 trials. As in the behavior-only task described above, 4 out of 6 fMRI participants had some blocks of trials where the stimuli were binned in 22.5˚ increments and ordered in a nonindependent manner (21 to 24 blocks/participant). However, all of the fMRI participants also participated in blocks with a uniform distribution of orientations across the entire 180˚ space (24 to 52 blocks/participant). For our main analysis, we include all trials from all participants. However, as with the behavioral analyses, we also report control analyses in which we repeat all of our main analyses excluding blocks with nonrandom stimuli.

## fMRI localizer task

Interleaved between the main task blocks, participants completed an independent localizer task used for voxel selection where they were presented with a sequence of grating stimuli at different orientations. Stimuli had a pseudo-randomly determined orientation that either matched the spatial location occupied by the *donut* stimuli used in our main task (outer diameter Ø = 21.2˚, inner diameter Ø = 2.37˚) or were a smaller foveal oriented Gabor corresponding to the "hole" in the donut stimuli (diameter Ø = 2.37˚). Participants were instructed to attend to 1 of 3 features orthogonal to orientation depending on the block: detect a contrast change across the entire stimulus, detect a small gray blob appearing over part of the stimulus,

or detect a small change in contrast at the fixation point. Each stimulus was presented for 6,000 ms and was separated by an ITI ranging from 3 to 8 seconds.

## Response bias

Each trial consisted of a stimulus and a probe separated by a probe offset ($\delta\theta$) that was either positive (probe is CW of stimulus) or negative. We report degrees in a compass-based coordinate system such that 0˚ is vertical and orientation values increase moving CW (e.g., 30˚ would point toward 1 o-clock). Participants judged whether the probe was CW or CCW relative to the remembered orientation by making a binary response. To quantify the precision and the response bias, we fit participant responses with a Gaussian cumulative density function with parameters μ and σ corresponding to the bias (mean) and standard deviation of the distribution. The likelihood of a given distribution was determined by the area under the curve (AUC) of the distribution of CW (CCW) offsets between the stimulus and the probe ($\delta\theta$) on trials where the participant responded CW (CCW). In extreme cases, a very low standard deviation (σ) value with no bias would mean that all $\delta\theta$ would lie outside the distribution and the participant would get every trial correct. A high negative bias (μ) value would mean that $\delta\theta$ would always lie CW relative to the distribution and the participant would respond CW on every trial. The best fitting parameters were found using a bounded minimization algorithm (limited memory BFGS) on the negative log likelihood of the resulting responses (excluded the small number of trials without a response) given the generated distribution [73]. We included a constant 25% guess rate in all model fits to ensure the likelihood of any response could never be 0 (critical for later modeling). While this was critical to fitting our model to raw data, the specific choice had no qualitative effect on our behavioral findings besides making the σ values smaller compared to having a 0% guess rate. By having a constant guess rate rather than varying it as a free parameter, we were able to directly compare σ values across participants as a measure of performance.

## Serial dependence

To quantify the dependence of responses on previous stimuli, we analyzed response bias and variance as a function of the difference in orientation between the previous and current orientation ($\Delta\theta = \theta_{n-1} - \theta_n$). We performed this analysis using a sliding window of 32˚, such that a bias centered on 16˚ would include all trials with a $\Delta\theta$ in the range [0˚, 32˚].

We additionally fit a Derivative of Gaussian (DoG) function to parameterize the bias of participant responses. The DoG function is parameterized with an amplitude A and width w

$$y = xAwce^{-(wx)^2},\qquad [1]$$

where $c = \sqrt{2e}$ is a normalization constant. For the purpose of fitting to our participant responses, x is $\Delta\theta$ and y corresponds to μ in our response model. For each participant, we adjusted 3 parameters: A, w, and σ to maximize the likelihood of participant responses. We report the magnitude of our fits as well as the resulting FWHM estimated numerically.

## Response precision

In addition to quantifying how responses were biased as a function of stimulus history, we also estimated how precise responses were depending on their unsigned distance from the previous stimulus ($|\Delta\theta|$). When quantifying variance difference between close and far trials, we "folded" trials with $\Delta\theta < 0$ so that the bias would generally point in the same direction and not artificially inflate our variance measure. Values from the bin with more samples (typically "far") were

resampled (31 repetitions) without replacement with the number of samples in the smaller bin and the median chosen to control for sample number differences.

## Scanning

fMRI task images were acquired over the course of four 2-hour sessions for each participant in a General Electric Discovery MR750 3.0T scanner at the UC San Diego Keck Center for Functional Magnetic Resonance Imaging. Functional echo-planar imaging (EPI) data were acquired using a Nova Medical 32-channel head coil (NMSC075-32- 3GE-MR750) and the Stanford Simultaneous Multi-Slice (SMS) EPI sequence (MUX EPI), with a multiband factor of 8 and 9 axial slices per band (total slices 72; 2-mm3 isotropic; 0-mm gap; matrix 104 x 104; field of view 20.8 cm; TR/TE 800/35 ms; flip angle 52˚; in-plane acceleration 1). Image reconstruction and un-aliasing was performed on cloud-based servers using reconstruction code from the Center for Neural Imaging at Stanford. The initial 16 repetition times (TRs) collected at sequence onset served as reference images required for the transformation from k-space to the image space. Two 17-second runs traversing k-space using forward and reverse phase-encoding directions were collected in the middle of each scanning session and were used to correct for distortions in EPI sequences using FSL top-up (FMRIB Software Library) for all runs in that session [74,75]. Reconstructed data were motion corrected and aligned to a common image. Voxel data from each run was de-trended (8TR filter) and z-scored.

We also acquired one additional high-resolution anatomical scan for each participant ($1 \times 1 \times 1$-mm3 voxel size; TR 8,136 ms; TE 3,172 ms; flip angle 8˚; 172 slices; 1-mm slice gap; $256 \times 192$-cm matrix size) during a separate retinotopic mapping session using an in vivo 8-channel head coil. This scan produced higher-quality contrast between gray and white matter and was used for segmentation, flattening, and visualizing retinotopic mapping data. The functional retinotopic mapping scanning was collected using the 32-channel coil described above and featured runs where participants viewed checkerboard gratings while responding to an orthogonal feature (transient contrast changes). Separate runs featured alternating vertical and horizontal bowtie stimuli; rotating wedges; and an expanding donut to generate retinotopic maps of the visual meridian, polar angle, and eccentricity, respectively [76]. These images were processed using FreeSurfer and FSL functions and visual ROI were manually drawn on surface reconstructions (for areas: V1-V3, V3AB, hV4, and IPS0).

## Voxel selection

To include only voxels that showed selectivity for the location of the oriented grating stimulus used in our main experimental task, we used responses evoked during the independent localizer task (see fMRI localizer task). For all analyses, we used TRs 5–11 (4 to 8.8 seconds) following stimulus onset. First, voxels were selected based on their response to the spatial location of the grating stimulus by performing a *t* test on the responses of each voxel evoked by the donut and the donut-hole stimuli, selecting the 50% of the voxels most selective to the donut for a given ROI. Of the voxels that passed this cutoff, we then performed an ANOVA across 10˚ orientation bins and selected the 50% of voxels with the largest F-score, thus retaining approximately 25% of the initial voxel pool. These selected voxels were used in all main analyses.

## Orientation decoding

We performed orientation decoding by training an IEM [77] on BOLD activation patterns using a sliding temporal window of 4 TRs. For most analyses, we focused on a 3.2 seconds (4 TR) window centered 6.4 seconds after stimulus presentation. We first designed an encoding model that assumes voxels are composed of populations of neurons with tuning functions

centered on 1 of 8 orientations evenly tiling the 180˚ space. The response of population i to stimulus θ is given by

$$c_i(\theta) = \max(0, cos^5(\theta - \omega_i)), \qquad [2]$$

where $\omega_i$ is the center of the tuning function. The response of voxel j is defined as a weighted sum of these hypothetical populations:

$$B_j = \sum_i^8 c_i w_i \qquad [3]$$

or in matrix notation,

$$B = CW \qquad [4]$$

where B (trial × voxel) is the resulting BOLD activity, C (trial × channel) is the hypothetical population response, and W (channel × voxel) is the weight matrix. The weight matrix W is estimated as

$$\widehat{W} = C^{-1}B \qquad [5]$$

where $C^{-1}$ (channel × trial) is the pseudo-inverse of C (implemented using the NumPy pinv function). We then estimated channel responses using the inverse of our estimated weight matrix:

$$\widehat{C} = B\widehat{W}^{-1} \qquad [6]$$

This channel response corresponds to a representation of orientation activity. To decode orientation, we took the inner product with a vector of the tuning curve centers in polar coordinates. The angle of the resulting vector was taken as the estimated orientation ($\widehat{\theta}$) while the vector length was taken as a proxy for model certainty ($\widehat{R}$).

$$\widehat{\theta} = angle(\widehat{C}e^{i\omega}) \qquad [7]$$

$$\widehat{R} = \|\widehat{C}e^{i\omega}\| \qquad [8]$$

The weight matrix of our model was estimated from a subset of our data and used to estimate orientation representations on a held-out portion of the task data. We used leave-one-block-out cross-validation where each block was a set of 4 consecutive runs (64 trials). These blocks had orientations that were linearly spaced across the entire 180˚, with a random phase offset for each block, to ensure a balanced training set. We performed an additional analysis training a model on all data from the localizer task and testing on the memory task. This model had lower SNR than models trained on the task but showed qualitatively similar results as our task trained neural decoder.

## Kernel-based decoding

**Estimating average voxel HRFs through deconvolution.** Because we are measuring the effects of previous stimuli on responses to the current stimulus, we did an additional analysis to quantify any influence of overlapping HRFs that last for 20 to 30 seconds (e.g., the "undershoot" that happens approximately 8 to 18 seconds poststimulus; see Fig 3A). To account for overlapping HRFs, we used deconvolution to estimate the average univariate response separately in each voxel in each ROI by modeling the responses to both the stimulus and probe for 30 TRs (24 seconds) poststimulus [44, 45]. We created a design matrix (rows × columns = total

number of TRs × 30) with the first column containing ones corresponding to the onset TR of each stimulus (and zeros elsewhere). Subsequent columns were the same vector shifted forward in time by one TR. Following the same procedure, another design matrix was defined for the probe onset times. These matrices were stacked with a column of ones added for each run as a constant term, yielding a final design matrix X of dimensions (total number of TRs × (60 +number of blocks)). We created a related matrix of voxel activity Y (total number of TRs × number of voxels) by concatenating responses in each voxel across blocks. We then estimated the HRF by performing least squares regression using the normal equation:

$$h = (X^T X)^{-1}(X^T Y) \tag{9}$$

The resulting weights corresponded to the average time course of the HRF evoked separately by the stimulus and the probe across all trials. We note that this HRF is estimated independent of the orientation of the presented stimuli as we wanted to use these estimates to then decode orientation dependent changes in activation patterns. For each voxel, we then parameterized the HRF using a 6-parameter double gamma function using scipy.optimize.minimize so that we could use the voxel-specific HRF model in a generalized linear model (GLM) to estimate the response magnitude in each voxel on each trial. We excluded the 11% of voxels which failed to converge on a solution.

**Estimating trial-by-trial responses using parameterized voxel HRFs.** For each voxel, we then created a design matrix Xv (rows × columns = total number of TRs × (number of trials * 2 +number of blocks)) with each column a delta function centered at the onset of the stimulus (or probe). We then regressed this matrix onto the (total # of TRs) vector Yv of voxel activity using Eq 9. This resulted in a simultaneous estimation of the trial-by-trial magnitude of responses to each stimulus grating and each probe which was repeated for each voxel to allow voxel specific HRFs to be utilized in the creation of Xv. The resulting activity pattern associated with each stimulus was used in the same manner as the raw time course of the BOLD response to train and test an IEM, and the resulting estimates should be largely independent of linear contributions of previous stimuli [44].

## Neural bias

To quantify how BOLD representations were biased by sensory history, we computed the circular mean of decoding errors ($\theta_{\text{error}} = \text{wrap}(\theta_{\text{decode}} - \theta_{\text{stim}})$):

$$\mu_{circ} = angle(\overrightarrow{R}), \tag{10}$$

$$\overrightarrow{R} = \frac{1}{nTrials}\sum_{k=0}^{nTrials} e^{i\theta_{error}^k}. \tag{11}$$

We estimated this bias using the same 32° sliding window as a function of Δθ used for visualizing response bias from participant responses. We additionally quantified the magnitude of the bias in decoding errors by fitting a DoG function to the raw decoding errors by minimizing the RSS and reporting the amplitude term.

## Neural variance

To quantify the variance of decoded orientations from visual areas, we computed the circular standard deviation on binned decoding errors:

$$\sigma_{circ} = \sqrt{-2\ln|\overrightarrow{R}|} \tag{12}$$

This was visualized using the same sliding window analysis as well as in reference to whether it was close or far from the previous stimulus.

## Dimensionality analysis

To quantify how stimulus history shaped the structure of neural responses independent of neural tuning we utilized principal component analysis (PCA). For a given set of neural responses R (number of trials × number of voxels) we mean centered and performed eigenvalue decomposition on the (number of voxels × number of voxels) covariance matrix. Eigenvalues were sorted in descending order and our response matrix was projected into PCA space (for visualization purposes) by multiplying by the sorted eigenvectors.

To compare dimensionality across conditions, we subset our data into trials following close (<30˚) or far (>60˚) trials and randomly sub selected trials from the larger group (without replacement) to equate trial numbers. We then performed PCA separately for each group and compared the relative proportion of total variance explained as the magnitude of the sorted eigenvalues. We quantified both the minimum number of components to reach at least 90% of the variance explained and also recorded the mean (AUC) of the variance curve.

## Modeling

We sought to develop a model that could explain both neural and behavioral biases as a function of stimulus history. For the fMRI data, we focused on explaining changes in encoding that could lead to the observed biases in the output of the BOLD decoder that was specifically designed to be "unaware" of stimulus history. To explain the behavioral data, we assumed that a decoder would receive inputs from the same population of sensory neurons that we measured with fMRI and that the decoder would read out this information in a manner that gives rise to attractive serial dependence. We considered readout models that were either unaware, aware, or overaware of adaptation and additionally applied a Bayesian inference stage, which integrates prior expectations of temporal stability, to the unaware and aware decoders [4]. We then compared performance between these competing models to see which could best explain our behavioral data.

Our full models consisted of 2 stages: an encoding stage where the gain of artificial neurons was changed as a function of the previous stimulus (adaptation) and a decoding stage where the readout from this adapted population was modified. The encoding population consisted of 100 neurons with von Mises tuning curves evenly tiling the 180˚ space. The expected unadapted population response is

$$Resp_N(\theta_n) = R\,\gamma_N e^{\kappa\,\cos(\Phi - \theta_n) - 1}, \tag{13}$$

where $\gamma_N$ is the scalar 1 for constant gain without adaptation, $\Phi$ is the vector of tuning curve centers, $\theta_n$ is the orientation of the current stimulus, $\kappa = 1.0$ is a constant controlling tuning width, and R is a general gain factor driving the average firing rate. We implemented sensory adaptation by adjusting the gain of tuning curves relative to the identity of the previous stimulus, $\theta_{n-1}$ (Fig 4A, Gain adaptation):

$$\gamma_A(\theta_{n-1}) = \gamma_N - rect\left(\gamma_m\cos^3(\gamma_s(\Phi - \theta_{n-1}))\right), \tag{14}$$

where $\gamma_m$ is the magnitude of adaptation, $\gamma_s$ scales the width of adaptation, and *rect* is the half-wave rectifying function. The responses of the adapted population thus depend on both the current and previous stimulus (Fig 4A, Efficient encoding):

$$Resp_A(\theta_n, \theta_{n-1}) = R\,\gamma_A e^{\kappa\,\cos(\Phi - \theta n) - 1}. \tag{15}$$

**Unaware decoder.** We first considered a model in which an adapted orientation-encoding representation is being decoded by an unaware readout mechanism (Fig 4B). The likelihood of each orientation giving rise to the observed response profile across $N$ neurons was estimated assuming activity was governed by a Poisson process:

$$P_{unaware}(Resp_A|\theta) = \exp(\sum_{i=1}^{N}\log P_{Poisson}(Resp_A^i(\theta); Resp_N^i(\theta))) \qquad [16]$$

$$P_{Poisson}(k; \lambda) = \frac{\lambda^k e^{-\lambda}}{k!}, \qquad [17]$$

where $Resp_N^i(\theta)$ is the expected response of the unadapted neuron i to stimulus θ and $P_{Poisson}(k; \lambda)$ is the probability of observing k spikes given an expected firing rate of λ. The decoded orientation is then the θ giving rise to the maximum likelihood estimation (MLE).

**Aware decoder.** In addition to the unaware decoder, we also evaluated the ability of a decoder that was aware of the current state of adaptation to explain behavior. The aware decoder differs from the unaware decoder in that its assumed activity level for each unit is modulated as a function of stimulus history:

$$P_{aware}(Resp_A|\theta_n; \theta_{n-1}) = \exp(\sum_{i=1}^{N}\log P_{Poisson}(Resp_A^i(\theta_n, \theta_{n-1}), Resp_A^i(\theta_n, \theta_{n-1}))). \qquad [18]$$

Note that here the rate parameter $k \equiv \lambda \equiv Resp_A$ such that the observed and expected values perfectly align with the presented orientation. $P_{aware}(Resp_A|\theta_n; \theta_{n-1})$ is dependent on sensory history and is nonbiased.

**Overaware decoder.** Our final decoding scheme we call the overaware decoder. This model can test whether serial dependence can be achieved without an explicit stage of Bayesian inference introduced in the next section. The decoder has an assumed adaptation defined by a unique set of free parameters, $\gamma_{m2}$ and $\gamma_{s2}$, which shapes a separate gain adaptation:

$$\gamma_{OA}(\theta_{n-1}) = \gamma_N - rect(\gamma_{m2} \cos^3(\gamma_{s2}(\Phi - \theta_{n-1}))), \qquad [19]$$

which, in turn, shapes the response profile of $Resp_{OA}$ in the same manner as $Resp_A$. The likelihood profile is then defined as

$$P_{over-aware}(Resp_A|\theta) = \exp(\sum_{i=1}^{N}\log P_{Poisson}(Resp_A^i(\theta); Resp_{OA}^i(\theta, \theta_{n-1}))), \qquad [20]$$

where our expected (assumed) rate λ is designated by $Resp_{OA}$. By having a larger assumed adaptation than implemented at encoding (through either $\gamma_{m2} > \gamma_m$ or $\gamma_{s2} > \gamma_s$) the net effect of the overaware decoder should be behavioral attraction.

**Bayesian inference.** In addition, we explored the effect of applying an explicit Bayesian prior based on temporal contiguity to the likelihood functions derived from these different readout schemes. This type of prior has been previously used to explain behavioral biases without considering how encoding might also be affected by stimulus history [4]. Specifically, the prior is defined by the transition probability between consecutive stimuli and is defined as a mixture model of a circular Gaussian and a uniform distribution:

$$P_T(\theta_n|\theta_{n-1}) = \frac{1}{Z} e^{-\frac{angle(\theta,\theta_{n-1})^2}{2\psi^2}} \qquad [21]$$

$$P_{Bayesian}(\theta_n|\theta_{n-1}) = P_{SAME}P_T(\theta|\theta_{n-1}) + \frac{1}{2\pi}(1 - P_{SAME}), \qquad [22]$$

with $P_{SAME}$ set to 0.64 (as found empirically in [4]), Z as a normalization constant so $P_T$ integrates to 1, and ψ is a free parameter describing the variance of the transition distribution.

This prior ([Fig 4C](), black line) is multiplied by the unaware likelihood ([Fig 4C](), yellow dashed line): to get the posterior estimate of our Bayesian-unaware decoder ([Fig 4C](), yellow solid line):

$$P_{Bayesian-unaware}(\theta_n | Resp_A; \ \theta_{n-1}) = P_{Bayesian}(\theta | \theta_{n-1}) P_{unaware}(Resp_A | \theta_n). \qquad [23]$$

We can additionally examine a Bayesian-aware decoder by substituting its respective likelihood function. We did not examine a Bayesian-overaware model so that all decoding models would have the same number of free parameters and so that we could directly evaluate the need for an explicit prior.

**Model fitting.** The encoding stage of the model has 2 free parameters and for each participant these parameters were optimized to minimize the RSS between our measured fMRI decoding errors and the decoding errors of our unaware decoder. For simplicity, we only fit our model to decoding errors from V3 as it had the highest SNR, but other early visual ROIs showed similar results. After fitting the encoding stage of the model, we then separately fit the 3 competing decoding models to best account for the behavioral data: Bayes-unaware, Bayes-aware, and overaware (2 free parameters each). The output of this readout stage was treated as the behavioral bias (μ), and the free parameters were optimized to maximize the likelihood of the observed responses (assuming constant standard deviation σ estimated empirically for each participant). For the purposes of fitting the model, the firing rates of the modeled neurons were deterministic (no noise process). Having noiseless activity had no effect on the expected bias (verified with additional simulations) and served to make model fitting more reliable and less computationally intensive. Both stages of the model were fit using the same cross-validation groups as our neural decoder. To ensure all models had a sufficient chance of achieving a good fit to behavioral data, we implemented a grid search sampling 30 values along the range of each variable explored (900 locations total) followed by a local search algorithm (Nelder–Mead) around the most successful grid point. We found dense sampling of the initial parameter space was especially important for our Bayes-unaware model.

**Model evaluation.** For bias of neural and behavioral responses, we evaluated the performance of the 2 stages of our model separately. These stages must be evaluated in a qualitatively different manner as the neural data give us an orientation estimate for each trial while the behavioral data consists of binary responses. For the encoding stage, we quantified how well the output of our unaware decoder predicted the raw errors of our BOLD decoder using circular correlation. The performance of this model was contrasted with the true presented orientation, which is analogous to the representation of an unadapted population. We additionally computed the variance of the neural decoding errors explained by the model bias ($R^2$). For the decoding stage of our model, we compared the log-likelihood of observed responses for each model.

We additionally estimated the variance of our models using neurons with rates generated by a Poisson process. The average bias was unaffected by allowing random fluctuations in activity, but the trial-to-trial variance increased. To get a stable estimate, we simulated 1,000 trials for each set of parameters estimated for a cross-validation loop for each participant and pooled these outputs. We compared the overall variance of our models to our single parameter estimate of participant precision using Jensen–Shannon divergence. We additionally examined relative precision of our model for close and far trials in the same manner as participant responses and decoding errors (Response precision).

## Supporting information

**S1 Fig. Response model.** Encoding of stimulus is assumed to be a noisy process whereby the distribution of encoded orientations is described by a Gaussian pdf with mean μ and standard

deviation σ. Dashed line is pdf, and solid line is the cdf of encoding distribution. Note that participants are reporting the probes orientation relative to the stimulus so more frequent CCW responses would correspond to a CW perceptual bias. **(A)** Example estimation curve with no bias and a very small σ. If the difficulty was set to δθ = 6˚ (3 sd) than this participant would get essentially all (99.7%) trials correct. **(B)** Estimation curve with a μ = −10, this participant would respond CW on almost every trial. **(C, D)** Realistic encoding curves. To aid with fitting and to best describe responses, a constant guess rate of 25% was included in the response model fit to participant responses. (C) An unbiased distribution with 2 theoretical stimuli on which the participant responded CW. The left response δθ = −6˚ is incorrect. (D) A CCW biased distribution results in a higher likelihood for all CW responses. Data and code supporting this figure found here: https://osf.io/e5xw8/?view_only= e7c1da85aa684cc8830aec8d74afdcb4. CCW, counterclockwise; CW, clockwise.
(TIF)

**S2 Fig. A subset of behavior only participants completed a version of the experiment with inhomogeneities in their stimulus sequences (such that consecutive orientations were not independent).** To confirm this manipulation did not drive any of our results, we repeated our behavioral analyses excluding participants with nonindependent sequences leaving a cohort of $n = 25$ with an average accuracy of 70.46 ± 1.14˚ at an average δθ of 4.97 ± 0.35˚. **(A, D)** This cohort still showed significant serial dependence (DoG amp = 4.71 ± 0.49, t(23) = 9.4, $p = 2.4^*10^{-9}$; width 0.027 ± 0.0019, FWHM 43.68 ± 1.86˚, **(B, C)** and had responses that were more accurate (t(24) = 3.14, $p = 0.0023$, **(E, F)** and precise following "close" stimuli (t(24) = −3.54, $p = 0.0009$, **(G)** Last, bias and variance were still positively correlated across this cohort (r(22) = 0.72, $p = 0.00003$, **(H–J)** Stimulus history effects are larger for worse performing participants. H: Serial dependence was significantly greater for less precise participants (t(45) = −2.5, $p = 0.012$, unpaired $t$ test comparing DoG Amplitude). (I–J) Variance was modulated significantly by stimulus history (low-performing: t(23) = 3.9 $p = 0.0007$; high-performing t(22) = 2.4, $p = 0.02$, 1-sample $t$ tests), with a significant interaction between overall performance and the effect size ($p = 0.017$, mixed effects linear model). Data and code supporting this figure found here: https://osf.io/e5xw8/?view_only=e7c1da85aa684cc8830aec8d74afdcb4. DoG, Derivative of Gaussian; FWHM, full width at half maximum.
(TIF)

**S3 Fig. A subset of fMRI participants completed some sessions where consecutive stimuli were not strictly independent. (A)** To confirm this structure was not driving our results, we repeated our main analyses excluding these sessions and found that responses were still strongly attracted to the previous stimulus (DoG Amp: 3.25 ± 0.34, t(5) = 8.85, $p = 1.53e-04$; DoG FWHM: 36.1 ± 2.9). **(B)** We found that responses were no longer significantly more precise following small changes in orientation but were trending in the same direction as when including all sessions (t(5) = −1.55, $p = 0.09$). We additionally confirmed that our finding of reduced bias around small changes in orientation was not driven by the oblique effect in the same manner as the behavioral cohort (mean % cardinal close: 48.6 ± 0.9%, far: 49.8 ± 0.2%, t(5) = −1.0, $p = 0.36$, paired $t$ test). **(C–E)** We further replicate our finding of neural repulsion and increased uncertainty following "close" stimuli across all ROIs except IPS0. **(F)** As a control analysis, we attempted but were unable to decode the identity of the next trial in any ROI when including all sequences. ns, not significant; *, $p < 0.05$; **, $p < 0.01$; ***, $p < 0.001$. Data and code supporting this figure found here: https://osf.io/e5xw8/?view_only= e7c1da85aa684cc8830aec8d74afdcb4. DoG, Derivative of Gaussian; fMRI, functional magnetic resonance imaging; FWHM, full width at half maximum; ROI, region of interest.
(TIF)

**S4 Fig. Impact of previous trial across time and individuals.** **(A)** Decoding of the previous stimulus dropped to chance around stimulus presentation before rebounding. **(B)** Decoding using sensory localizer data was consistently at chance during N+1 trial suggesting information relating to past stimulus is not stored in a sensory code. **(C, D)** Decoded biases across time for both decoders are consistently repulsive. **(E)** Bias curves for individual participants using the memory decoder across ROIs (see legend) overlayed with behavioral biases (black). Neural and behavioral biases are consistently in opposite directions. Note that id#3 exhibits peripheral repulsion. Data and code supporting this figure found here: https://osf.io/e5xw8/?view_only=e7c1da85aa684cc8830aec8d74afdcb4. ROI, region of interest.
(TIF)

**S5 Fig. To quantify the intrinsic dimensionality of neural representations and whether it changes following a "close" stimulus, we performed PCAs on the activity matrix (number of trials × number of voxels) of responses across different ROIs.** **(A)** We found that early principal components were correlated with the presented orientation, here presenting both individual trials as well as the average location for different orientation bins (large solid circles) for an example participant and ROI. **(B)** We performed PCA separately for trials following "close" and "far" trials, being careful to subsample the number of trials in the larger group. We then sorted the eigenvalues and examined the proportion of variance explained as a function of the number of components included separately for each group. **(C)** We found that it took significantly more components to explain 90% of the variance on the population activity following close versus far stimuli. This suggests that the representations in most visual areas occupy a higher dimensional space following close stimuli, but curiously not V1. Note that the total number of dimensions is shaped by the number of voxels included, so differences between participants/ROIs should not be interpreted with how these data were processed. **(D)** We additionally looked at the area under the variance curve to avoid any arbitrary effects of choosing 90% and found a similar effect (higher AUC implies lower dimensionality). Data and code supporting this figure found here: https://osf.io/e5xw8/?view_only=e7c1da85aa684cc8830aec8d74afdcb4. AUC, area under the curve; PCA, principal component analysis; ROI, region of interest.
(TIF)

**S6 Fig. Decoded uncertainty as a function of Δθ across ROIs.** **(A)** $\sigma_{circ}$ of decoding errors is significantly greater for close ($<30°$) versus far ($>30°$) stimuli across early visual ROIs (see Neural variance). Points and error bars are mean ± SEM across participants; gray lines depict individual participants. Error bars depict SEM across participants. **(B)** Sliding $\sigma_{circ}$ for V1-V3 shows a monotonic relationship (± SEM across participants). **(C, D)** Same as A and B but measuring uncertainty directly measured from the single trial posterior (see Eq 8). Results are qualitatively very similar for both techniques. *, $p < 0.05$, **, $p < 0.01$, ***, $p < 0.001$. Data and code supporting this figure found here: https://osf.io/e5xw8/?view_only=e7c1da85aa684cc8830aec8d74afdcb4. ROI, region of interest.
(TIF)

**S7 Fig. To better understand how our experiment's trial sequence could impact results, we simulated BOLD signals based on our empirically estimated HRFs and our trial sequences used in the task.** We first created a population of 32 voxels with uniformly distributed von Mises tuning curves. Note that for the purposes of this simulation, we are effectively treating voxels as neurons instead of a summation of the metabolic demands of many neurons. This shortcut comes from experience simulating voxel activity and finding decoding results are

unaffected by such a shortcut while making results a bit simpler to understand (and faster to generate). The responses of each voxel were estimated by first generating a design vector based on the stimulus presentation times of both the stimulus and probe for a given participant with the amplitude of the response based on the defined tuning curves. This vector was then convolved with an empirically estimated HRF (both the raw output and when parameterized with a double gamma function) randomly sampled from voxels of the same participant to get the estimated evoked response to both the stimulus and the probe. These 2 signals were then combined along with gaussian noise to simulate the voxel response (**A**). Importantly, the tuning properties of these simulated voxels were unaffected by past stimuli so any biases found by applying our decoding techniques could reflect artifacts of our task design or analysis procedure. We additionally simulated BOLD responses with true adaptation in the underlying neural tuning. For simplicity, we simply attenuated the response to the current trial by 40% of the response to the previous trial while keeping all other stages of our analysis the same. We first applied a decoder across time to the epoched data and found a similar pattern to our empirical data with decoding performance following a parabolic shape before leveling off at some intermediate level, here utilizing HRFs from V3 voxels (**B**). This was true whether we used parameterized or raw HRFs and whether the simulation included adaptation. We next examined biases in our decoder as a function of stimulus history. With adaptation (red curves), decoded representation were systematically repelled from previous stimuli matching our empirical findings (**C**). Importantly, without adaptation the resulting bias was never repelled from the previous stimulus (blue curves). This suggests that the timing of our stimuli and the resulting evoked responses should not bias us toward seeing the repulsive results we report. We finally implemented the regression-based estimation of BOLD responses as we did with our empirical data. As stated before, this technique should remove any linear contributions of past evoked responses to our estimate of the current trial's response. When analyzing the resulting biases, we found that while the unadapted data showed no bias from the previous stimulus (as expected, despite added noise) the adapted response continued to show a repulsive bias (**D**). This analysis demonstrates that (1) while our task design could lead to biases in decoded representations in the absence of any neural history effects, these effects tend to be in the opposite direction of our reported effects and (2) our use of HRF kernels to estimate trial responses is unbiased by across trial contamination and robustly recovers repulsive patterns in the presence of real neuronal adaptation at noise levels similar to our study. Data and code supporting this figure found here: https://osf.io/e5xw8/?view_only=e7c1da85aa684cc8830aec8d74afdcb4. HRF, hemodynamic response function.
(TIF)

**S8 Fig. Model fits for individual participants (same order as Fig 3).** Solid lines correspond to empirical neural (yellow) or behavioral (green) bias; dashed lines correspond to model fits to BOLD decoding bias (Unaware model, **A**) or behavior (**B–D**). Model fits plotted are average of noiseless biases generated by models fit to each CV fold. Note that models are fit to raw data, not binned data presented here. Pearson correlations are reported above each fit between binned and model estimated bias. Data and code supporting this figure found here: https://osf.io/e5xw8/?view_only=e7c1da85aa684cc8830aec8d74afdcb4.
(TIF)

**S1 Table. Cells correspond to parameters for proposed decoder.** Items with bold values indicate free parameters adjusted to fit empirical data (± SEM across participants). $\gamma_m$ controls the amplitude, and $\gamma_s$ controls the width of gain adaptation (Fig 4A). These parameters were fit by minimizing the RSS between the unaware decoder and the BOLD decoder output. $\gamma_{m2}$ and $\gamma_{s2}$ are the assumed adaptation parameters at decoding. These terms were either set to assume no

adaptation (unaware), match the true amount of adaptation (aware) or are free parameters adjusted to maximize the likelihood of responses (overaware, Fig 4B). Last, R adjusts the average Poisson firing rate and ψ controls the variance of the prior distribution (Fig 4C). These parameters are adjusted for decoders using a Bayesian prior while R is set to the arbitrary value of 5 for non-Bayesian decoders (it has no effect on bias for non-Bayesian decoders). Increasing R increases the precision of the likelihood function and reduces the relative influence of the prior. Increasing ψ increases the range of Δθ over which the prior has an influence. Data and code supporting this figure found here: https://osf.io/e5xw8/?view_only= e7c1da85aa684cc8830aec8d74afdcb4. RSS, residual sum of squared errors. (DOCX)

## Acknowledgments

Thanks to Chaipat Chunharas for critical discussions in experimental design and assistance with scanning and to Anika Jollorina and Shuangquan Feng for assistance with behavioral data collection. Thanks to Marcelo Mattar for helpful comments on our model; to Gal Mishne for help with setting up dimensionality analyses; and to Margaret Henderson, Sunyoung Park, and Kirsten Adam for thoughtful comments on earlier versions of the manuscript.

## Author Contributions

**Conceptualization:** Timothy C. Sheehan, John T. Serences.

**Data curation:** Timothy C. Sheehan.

**Formal analysis:** Timothy C. Sheehan.

**Funding acquisition:** John T. Serences.

**Resources:** John T. Serences.

**Software:** Timothy C. Sheehan.

**Visualization:** Timothy C. Sheehan.

**Writing – original draft:** Timothy C. Sheehan.

**Writing – review & editing:** Timothy C. Sheehan, John T. Serences.

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
