## [Editor Report · Decision Letter 0]

19 Nov 2021

Dear Dr Sheehan, 

Thank you for submitting your manuscript entitled "Sensory readout accounts for adaptation" for consideration as a Research Article by PLOS Biology.

Your manuscript has now been evaluated by the PLOS Biology editorial staff, as well as by an academic editor with relevant expertise, and I am writing to let you know that we would like to send your submission out for external peer review.

Once your full submission is complete, your paper will undergo a series of checks in preparation for peer review. Once your manuscript has passed the checks it will be sent out for review. 

If your manuscript has been previously reviewed at another journal, PLOS Biology is willing to work with those reviews in order to avoid re-starting the process. Submission of the previous reviews is entirely optional and our ability to use them effectively will depend on the willingness of the previous journal to confirm the content of the reports and share the reviewer identities. Please note that we reserve the right to invite additional reviewers if we consider that additional/independent reviewers are needed, although we aim to avoid this as far as possible. In our experience, working with previous reviews does save time. 

If you would like to send your previous reviewer reports to us, please specify this in the cover letter, mentioning the name of the previous journal and the manuscript ID the study was given, and include a point-by-point response to reviewers that details how you have or plan to address the reviewers' concerns. Please contact me at the email that can be found below my signature if you have questions. 

Please re-submit your manuscript within two working days, i.e. by Nov 23 2021 11:59PM.

Kind regards,

Gabriel

Gabriel Gasque

Senior Editor

PLOS Biology

ggasque@plos.org

---

## [Decision Letter · Decision Letter 1]

24 Jan 2022

Dear Dr Sheehan,

Thank you for submitting your manuscript "Sensory readout accounts for adaptation" for consideration as a Research Article at PLOS Biology. Your manuscript has been evaluated by the PLOS Biology editors, by an Academic Editor with relevant expertise, and by three independent reviewers. Please accept my apologies for the long delay in communicating this decision to you.

In light of the reviews (below), we will not be able to accept the current version of the manuscript, but we would welcome re-submission of a much-revised version that takes into account the reviewers' comments. We cannot make any decision about publication until we have seen the revised manuscript and your response to the reviewers' comments. Your revised manuscript is also likely to be sent for further evaluation by the reviewers.

We expect to receive your revised manuscript within 3 months. 

**IMPORTANT - SUBMITTING YOUR REVISION**

*Re-submission Checklist*

*Published Peer Review*

*PLOS Data Policy*

*Blot and Gel Data Policy*

Sincerely,

Gabriel

Gabriel Gasque

Senior Editor

PLOS Biology

ggasque@plos.org

REVIEWS:

Reviewer #1: In this paper, Sheehan and Serences explore very convincingly the neural correlates of serial dependence in the visual cortex using fMRI. What is more, they use Bayesian modelling to back the proposal that while there is repulsion at the sensory level, downstream readouts can compensate for this repulsion, leading to the typical attraction. Finally, they (somewhat) replicate their findings in the mouse visual cortex. 

I think this paper is of great interest to the cognitive neuroscience community, in particular those interested in serial dependence. I support the publication of this work at Plos Biology and I have some recommendations that I believe could improve the impact of the paper. 

Overall, I think the paper is rather long, e.g. with control analyses given a full figure. My feeling is that this dilutes the major findings of the paper. I would go through it and reconsider moving each figure/panel to supplementary material (currently empty). Some obvious examples are Fig 3, 9. My personal experience was that I was rather tired towards the end of the paper, which is a pity since the model is a crucial component of the paper. The modeling par itself could be compressed in just one figure with clearer messages 

A more important concern is the choice of folding the serial dependence curves (as proposed in Barbosa and Compte 2020, not Barbosa et al. 2020; line 98) and *then* unfolding it again. The choice of folding serial dependence curves to improve power and remove systematic biases indeed assumes the curves are symmetric, but copy-pasting it back seems to be going too far. Relatedly, I would add to the supplementary material the unfolded curves. 

I think it is also important to discuss Hajonides et al paper (https://doi.org/10.1101/2021.10.31.466639). While I acknowledge that the current study was uploaded to biorxiv before, and has several major advantages, that paper is closely related with the current study and should be discussed for this submission. 

Below, I provide some more specific comments or suggestions that I recover as I go through the paper once again, thus not ordered by importance. I want to state clearly that these are just suggestions or ideas that popped into my mind as I read the paper and should not be taken as a case for acceptance/rejection.

Line 64. "Further, existing data from early visual " - which data? Please cite. 

84-86. What correlates more with serial dependence, accuracy (as shown in the paper) or \\delta \\theta? 

123-. At this point I wondered what would be lost if only subjects with independent stimuli were considered for this study. If the results are upheld as it is claimed, I would remove completely the subjects with dependent stimuli. This would simplify the reading and length of the paper. If not, I would add a supplementary figure showing the full figure(s) only with/without those.

138. This is a weird finding, as many other studies have shown dependency on delay (Bliss et al 2017, Stein et al. 2020). I think this deserves a bit more unpacking / speculation about why you think you did not replicate this classical finding

Figure 2. Why not include the fixation period in the decoding analyses? This would be particularly interesting when decoding previous-trial stimuli as in Baes, Barbosa, Papadimitriou. I suggest moving panel E to supplementary and adding the time course of previous trial decoding, in particular during the ITI. 

148-150. Is this a one sided t test? It should be. Actually, given that you found a significant finding outside the scanner, the threshold should not be 0.05 (even if one sided). Consider using other analyses such as the Bayes factor. 

173. This is confusing and it seems very important for the interpretation of the fmri results. I would clarify this better. 

182. \\mu_circ was not introduced. 

183. These neural serial dependence curves are beautiful. Curiosity: how would the serial dependence curves look if fully built from neural predictions? If I understand correctly, only the y axis is from neural data. 

Figure 5 (and elsewhere) I would use + for future and - for previous. 

Figure 6. I would consider adding a schematic illustrating the task design. Why is C so different from B?

264. It is surprising the p value is 0.049. Judging by the figures the effect is so strong...

267. Why use a deconvolution kernel if it introduces spurious serial dependence with future trials? The origin of these future trials needs to be clarified, IMO. For example repeating the analyses without the deconvolution. And/or showing that it does not appear for purely behavioral plots as in Fig 5F.

Side note: In the current format, it seems the paper could finish right here. It is really challenging to embark on a completely different route (the Bayesian model). While I think this modeling work is crucial, I would consider moving most of it to supplementary and keeping only the main results/message. Note that in line 441 you summarize what is learned from the model. Going back to the figure, I feel it is possible to compress them substantially. 

291. m is undefined at this point.

299. Not clear what a 'formal prior based on temporal contiguity' is

Figure 8. Percept is undefined. More importantly, my intuition is that there is some circularity in the reasoning done in panel B. I would explicitly explain why there isn't. 

427. This is a very interesting point. I wonder if this could be quantified in the data (perhaps in future projects). It occurs to me to study dimensionality (e.g. with PCA) for different task conditions or subjects. Can one find a relationship between serial dependence and the dimensionality of neural representations? 

432-433. I would cite these two papers separately. Barbosa et al 2020 is another possible citation here. 

480. This is a very interesting point. I would unpack it more in the results and abstract. 

Finally, while reading both Hajonides and the current paper, it occurred to me that the repulsive neural correlates could be related to repulsive serial dependence seen when there is no current-trial delay (Bliss et al 2017). In general, I think it is good practice to have a delay-0 control condition when studying working memory. My prediction is that the neural repulsion seen here would correlate with the delay-0 repulsion. Unfortunately, this condition was not present in the fmri experiments and *I am not* suggesting new fmri data should be collected. However, please consider the following. Is there a correlation between attractive serial dependence for the protocols used inside and outside the scanner? If that is the case, one possibility to falsify my hypothesis would be to collect new, outside the scanner data, but including a delay 0 condition. Does that repulsion correlate with the neural repulsion? Again, this is pure speculation based on my own curiosity and I would not support a rejection case based on this 'missing' data.

Reviewer #2: A set of behavioral and fMRI experiments are presented on the neural basis of orientation serial dependence. This manuscript has many strengths: it uses sophisticated methods and presents data across species, even including reanalysis of mouse two-photon imaging data. It's an excellent manuscript in many ways, and it's very impressive too. My review is long, but that is because the manuscript is simultaneously excellent and yet the writing needs substantial revision. The paper is currently framed with the central question, essentially, "is serial dependence in visual cortex or outside visual cortex." This is a strawman. It would be better to reframe the paper to ask the unbiased but more important and open-ended general question: "Where can we find a manifestation of serial dependence in orientation". The results support a model in which there is a manifestation of orientation serial dependence in the read-out of information (wherever and however that happens). That is an interesting and novel idea, and it should be shared with the community. However, the discussion and conclusions about perception/decision/memory are not justified and the results are overinterpreted. That's easily fixed, though, because the manuscript doesn't test memory versus perception and doesn't need to be couched in the black-and-white debate about perception versus memory (just rephrase the question, as suggested above). The particular task and design (delay, working memory necessity, high contrast stimulus, full field presentation, etc, etc) favors finding an involvement of memory, favors producing adaptation, and could wash out any serially dependent pattern of activity in visual cortex, even if it were present. The null result—that visual cortex isn't involved--is hard to prove. As Kiyonaga et al, (TiCS, 2017) suggested, and as we now know from many subsequent papers, there are many forms of serial dependence, including in perception (appearance) and decision and memory (and in other domains like action too). There is also stimulus specific serial dependence (numerosity, face expression, attractiveness, timing, even audition and olfaction, etc etc). The experiments in this manuscript are interesting and worth reporting, but they need to be motivated and discussed in a more balanced way. With the revisions suggested below, I expect this will be a highly valuable paper that is certainly of broad interest to readers of PLoS Biology.

1. Abstract. Rephrase the sentence "Serial dependence is not implemented in visual cortex but rather by readout mechanisms that account for adaptation during encoding" to read "serial dependence is implemented in the readout mechanisms from visual cortex that account for adaptation during encoding". It is too strong (and loaded) to say that serial dependence is not "in" visual cortex (whatever that might mean). Who knows where the readout mechanisms are located, whether there is feedback, and whether the techniques here are able to isolate that readout is far from clear given that the mechanisms of readout itself are not fully specified or understood. 

2. The introduction introduces the terms "encoding" and 'readout': "We use the term "encoding" to refer to the initial conversion of external sensory information into neural activity patterns and the term 'readout' to refer to the readout of these encoded signals to shape behavior." These definitions are problematic and confusing. The "initial conversion of external sensory information" sounds more like transduction at the retina. Surely, there are "patterns" of responses in the retina and there is plenty of "neural activity patterns" in LGN and other places before V1 (not to mention feedback). These definitions just do not make sense, even loosely or operationally. 

3. Trying to define the "initial" or first "conversion" is not possible in this experiment and with these data. Trying to limit "encoding" to any particular level is also impossible. Encoding and read-out/decoding could happen at many levels and in many interacting ways. It seems that part of the problem is with the linear, hierarchical, and bottom-up way of thinking about visual processing, and (most problematic) the desire to simplify a straw man version of serial dependence as being either "in V1" or "not in V1". These are obviously oversimplifications. 

4. The abstract states that, "Previous behavioral studies suggest that serial dependence is implemented via modulations in visual cortex." In contrast to that quote, it is well-known in the literature that serial dependence requires attention (e.g., Fischer & Whitney, 2014; Kim, Burr, et al, 2020; Fritsche, 2019; Rafiei, et al., 2021; etc) and it occurs over a spatially broad region—far larger than any V1 RF (e.g., Fischer & Whitney, 2014; Fritsche, et al., 2017; Collins, Sci Rep, 2019; Cicchini et al., 2017; Manassi, Kristjansson, et al, 2019; etc). So, the idea that feedback (e.g., Cicchini et al., Curr Bio, 2021) and that more than V1 must be involved is not actually new (the only paper to really suggest V1, per se, was St. John-Saaltink et al). In fact, even Fischer's 2014 paper never mentions V1 or visual cortex, and the labeled line models they do mention could be implemented in the "read out" in a manner very similar to what is proposed in the current manuscript. To suggest--or imply--that the current results overturn a lot of psychophysical work is simply not justified. This should be acknowledged. Rather than framing the paper as a black-and-white "V1" versus "not-V1" the paper could incorporate a discussion of the richer and more nuanced prior psychophysical work. 

4. The authors use a delayed discrimination task. This design and approach are fine. However, other approaches also work to generate serial dependence and do not depend on delay or working memory. For example, there is serial dependence even in detection tasks (e.g., Murai & Whitney, Curr Bio, 2021) and simultaneous comparisons (e.g., Cicchini et al, JoV, 2017), among many other techniques (ratings, magnitude estimation, method of single stimuli, adjustment, etc etc). The delayed working memory dependent discrimination task is therefore not necessary to generate serial dependence, but it does favor finding something that hinges on that delay, and this risks biasing the results toward a memory-dependent form of serial dependence. That is perfectly fine—there is serial dependence that hinges on memory (e.g., Bliss et al., 2018)—but this does not rule out other forms of serial dependence. And, we do not know what would happen in visual cortex if another design were used.

5. How do we know that serial dependence "manifested in V1" was not simply washed out by a stronger negative aftereffect? Several papers have demonstrated that the same stimuli can generate both positive and negative aftereffects (depending on things like duration, contrast, size, noise, etc), and that they can be additive. This even goes back to Fischer 2014, as the authors acknowledge, who found negative aftereffects for either long adapting durations or ignored stimuli. Many other papers have found additive positive and negative aftereffects as well. How can we rule out the possibility of superimposed aftereffects that are just dominated by one or the other in this particular experiment?

6. The method of estimating an effective PSE is novel but somewhat dense and confusing. Although the authors took some pains to try to explain how "response bias" relates to "behavioral bias," the relationship is still obscure. What is estimated response error as a function of orientation (not difference in orientation)? On first blush it appears that the response bias is a repulsion, but that's not clear and is easily misinterpreted. How do Fig1B-C relate to each other? Is panel B the best way to convey the steps in the analysis to get to panel C? 

7. There is a concern about mirror flipping the data because the stimulus distribution is already circular and it's not clear why mirror flipping or merging data should be necessary unless there is some other systematic response bias lurking in the data (e.g., consistent biases reporting clockwise, response bias toward cardinal orientations, repulsion away from cardinal orientations, etc etc). Flipping and merging can cause something that is not serial dependence to look like serial dependence (e.g., central tendency biases). That a previous study did it is not very convincing justification (and some previous papers did this kind of thing for linear dimensions, mistaking a central tendency bias for serial dependence). Perhaps the raw non-mirrored data can be presented in supplemental material? 

8. How do we know the "read-out" does not happen in signals directly from V1? How do we know where the "read-out" happens? 

9. P. 4. The proposed distinction between serial dependence as "sensory" versus "working memory" is an oversimplified strawman. As Kiyonaga et al pointed out in what is now an old paper (2017, TiCS), there could be many forms of serial dependence (memory, perception, action, etc), and the notion of serial dependence as being black and white "either/or" is simply not supported in the literature. There are ample demonstrations of serial dependence without working memory (e.g., Cicchini et al., 2017; Murai & Whitney, 2021; Liberman et al., 2014), and also serial dependence that does depend on working memory (Papadimitriou et al, 2015; Bliss et al., 2017; etc). The mechanisms of adaptation and serial dependence need not be restricted to just one processing stage (e.g., see Collins, JEP, 2021; Liberman, et al., 2014, 2018; Taubert, Alais, & Burr, 2016; etc), and they may depend on stimuli and task. Serial dependence does not just happen for orientation, it happens for numerosity, face expression, identity, object shape, and other stimulus dimensions. I'm not aware of any paper that suggests that all of these serial dependencies are the same or that they all occur in V1. 

10. Fig. 2F. Decoding the previous stimulus replicates prior work and is useful. Can you try the same analylsis on the N+1 stimulus (future)? It shouldn't work, so it's a good confirmation that other artifacts aren't lurking.

11. Fig 2H. The direction of the effect is a bit confusing because it doesn't seem to match the caption description ("Decoded variance is significantly less.." and yet it is opposite panel 2B). 

12. Fig. 6D X axis labels confusing. Positive and negative numbers are confusing when using the words "trials back". What is the n-1 trial? Is that the positive 1 "trial back" or is that the -1 trial back (with a double negative)? Try using "back" and "forward" or some other way to disambiguate what is meant.

13. The large, noise-free, high contrast stimulus could bias the results toward finding a negative aftereffect in visual cortex (sensory adaptation). When contrast is increased and noise reduced, serial dependence tends to be weaker (e.g., Cicchini, et al; see also Kim & Alais, Vis Res, 2021), and if there are additive or separable effects in V1, the serial dependence might not show up. The flickering (phase reversal every 250 msec) could have a similar impact, improving measurement of adaptation and diminishing power to measure a neural signature of serial dependence. The long duration (compared to behavioral studies) of the stimulus (1 sec) could have a similar effect. 

14. The point above casts doubts on the generality of statements like, "This repulsive pattern suggests that serial dependence is not a direct result of attractive biases in early sensory areas." Statements like that are not more valid than other conclusions based on the inconsistency of the measured visual cortex responses with perception. For example, the wildly strong adaptation (repulsion) effects (Fig 3C) in early visual areas (perhaps helped by high contrast, full field stimuli) also do not produce a negative perceptual (tilt) aftereffect, but we would hesitate to conclude from these data alone that "V1 has no direct influence on perception of orientation." Maybe that is true and maybe it isn't. What we can take away is that the neural patterns definitely do not match behavior. 

15. Fig. 3. The variability in the "neural bias" between subjects is enormous and the source of it is unclear but is concerning. Some observers have around 7 deg peak bias and some observers have well over 15 deg, and one has well over 30 deg of bias. Despite that variability, the behavioral effect seems relatively consistent across participants. 

16. p. 32. "For some behavioral participants (n=9) delay and ITI were varied between 0.5-7.5s without notable effects on performance." This result seems at odds with the papers cited on page 4. Any ideas why? 

17. Fig. 5. It is not clear what, if anything, we can take away from the very long duration repulsive effects. They have little or no tuning, they don't match behavior (null result) and the conclusion from that figure was that "Together this suggests that adaptive biases may be much stronger and more persistent than previously thought." Previous behavioral studies have reported long term negative and positive aftereffects (e.g., Gekas, et al., JoV, 2019). However, the results here do not speak to that. 

18. "This suggests that the adaptive biases are more persistent than often thought and provide further support for a neural basis of our findings." Again, this is a stretch and a very tenuous connection. Fig 5 shows a repulsion that lasts 52+ sec (or more, since only 3-back was reported) and that is far longer than 10 sec in the cited paper. It's not comparable and whether an artifact causes the long duration suppression in the fMRI data remains unclear. Additionally, there are many previous behavioral studies on the time course of serial dependence, but the time course of the neural repulsion here doesn't agree with those. 

19. Fig 6. Apparently, the most significant effect (n+1 positive effect) is actually an artifact. That's a concern and the explanation for it is hand waving and reveals more about a problem with the experiment design and analysis. Although most of that wasn't under the authors' control, whether to present it and how to reanalyze and interpret it is. 

20A. Fig. 9. The model gets close to making predictions about individual differences, which is an excellent goal. However, the actual predictions for the behavioral data (e.g., Fig 9B or 9D) are nearly identical across observers and rather untethered from the neural (yellow) data. That suggests different observers predict each other pretty well. Whether individual differences are captured in any way is unclear (granted that 6 subjects is too few for any individual differences, but there should be some observer specific correspondence; the change in variability between observers is unclear). 

20B. How is the "neural" data in Fig 9A related to the "neural" data in Fig 3C? Shouldn't they match or correspond closely? Some observers do appear to have similar or corresponding data (ID#5) and some don't (ID#4 looks way off), but it is unclear why.

20C. Fig 9. The model works here, but the time course in the neural data is far off what has been reported in the literature (and far off that found in the behavioral data here) so whether the model would generalize is unclear. 

20D. Whether the results would hold in other more "perceptual" tasks (working memory is not required for serial dependence) is also unclear but has not been tested. That limit should be acknowledged.

21. Discussion section. "More generally, our BOLD data argue against an early sensory or 'perceptual' account of serial dependence and instead suggest that serial dependence is driven by post-perceptual or mnemonic circuits." This is not valid. The referenced "perceptual" account does not and need not make claims about V1 or other cortical areas being the seat of perception or the source of serial dependence. Those papers are psychophysical: the "perceptual" account is that "perception" is impacted by serial dependence. The current manuscript does not speak to that debate at all. The primary question in this manuscript was, as stated above, "Where can we find a manifestation of [orientation] serial dependence?" The authors do succeed in proposing a reasonable and interesting answer to that question—the read out of information might underlie at least one type of serial dependence—but, the authors have not tested anything about perception (e.g., appearance) versus decision versus memory (independent of perception). The sentence should be deleted and the rest of the discussion rebalanced to reflect the much richer literature, including the references in this review (listed below).

22. The perceptual account of serial dependence is a psychophysically based theory (or set of theories: Cicchini et al; Liberman et al; Collins; Alais et al; Fischer et al; Fornaciai et al; etc), it is not that V1 is responsible for serial dependence (of orientation, numerosity, faces, or anything else). The debate about whether serial dependence is in perception and/or decision and/or memory (see Kiyonaga et al, 2017) is not addressed by invoking V1 or ruling it out (or ruling out any other cortical area, for that matter). Whether serial dependence is "in" V1 or in the "read-out" from an area like V1 does not address or disprove the substantial behavioral literature. 

23. General comment. Serial dependence has been reported in many dozens of papers and in many domains, from numerosity (Corbett et al; Cicchini et al; Fornaciai et al; etc), orientation, face expression (Liberman et al., 2018; Alais et al., 2021; etc), face identity (Kok et al., 2017; Turbett et al., 2019; Liberman, et al, 2014; etc), shape (Manassi et al., 2021), texture (Manassi et al., 2017), attractiveness (Taubert et al, 2016; Taubert & Alais, 2016; Van der Burg et al., 2019; Xia et al, 2016; etc), to timing, audition (Ho et al., 2019), cross modal perception, and now even smell (Van der Berg, et al., 2021). Several papers have reported dissociations between these dimensions (e.g., Fornaciai & Park, 2019; Collins, 2021; others), showing that serial dependence is not monolithic; there is not a single homunculus or module for serial dependence—it is domain/stimulus/task specific. This manuscript only tests orientation, and the results only apply for orientation. Throughout the manuscript, the text should be specific that the experiments, results, and conclusions are only about orientation serial dependence, and the result may not generalize to any other stimulus.

24. p. 26. "However, others have found that serial dependence is repulsive at very short delays and only becomes attractive when items are held for an extended time in working memory (Papadimitriou et al., 2015; Bliss et al., 2017)." These cited papers are about stimulus position, not orientation, and do not hold for the current experiment. First, the authors own data (delay, ITI dependence) doesn't seem to support this statement (and if it does, that needs to be reported). Second, other papers have not found these delay effects for orientation (or other stimuli). Third, if the authors want to rely on unrelated (non-orientation) stimuli for their preferred interpretation (selectively citing Bliss and Papadimitriou, for example), then they need to be thorough and balanced about citing papers on serial dependence across many domains (shape, faces, numerosity, tone, timing, etc etc). 

25. p. 26. "Consistent with the perceptual account, an fMRI study with low contrast stimuli… consistent with a perceptual origin" The cited paper is not "consistent" or "inconsistent" with the perceptual account. It didn't psychophysically measure anything about perception and didn't tie the results to perception (it also suffered from the problems the authors describe at length). It suggested V1 as a possible correlate of orientation serial dependence, and the current results disagree with that. But, the current results do not speak to perception vs memory vs decision. It is not justified to equate V1 to "perception": the "perceptual account" of serial dependence is not the same thing as the "V1 account" of serial dependence. This confusion permeates the manuscript. The results in this manuscript might reject a V1 account (V1 as the sole source of serial dependence) but it does not test or reject the "perceptual account[s]". 

26. p. 26. "Thus, while some behavioral studies show convincing evidence of serial dependence emerging immediately with perception, the existing neuronal evidence supporting this account is inconclusive." It is true that there is relatively little neurophysiological evidence for the origins of serial dependence in any stimulus domain (despite the fact that it is found in hundreds of behavioral papers on all manner of stimuli). There are several uncited papers, however, that have investigated possible neural sources (e.g. using EEG; Fornaciai & Park, 2018, 2020; Bae, Cer Cor Comm, 2021; and MEG too, etc) and some of these papers do find early signatures of serial dependence. The real problem here is that being "immediate" is not required for "perception". It is not the immediacy of the effect that matters, it's the fact that perception, per se, is impacted. "Perception," for example, is operationalized in some previous studies as "appearance" (e.g., Collins, 2020; Cicchini, et al., 2017). If the authors want to redefine "perceptual" as something else, they need to justify that, but it's not valid to simply equate "perceptual" to "V1". 

27. "Thus, in line with our findings, behavioral and neuronal studies using high contrast stimuli have found results consistent with a post-perceptual account of serial dependence." Again, this misleadingly equates V1 (or some early cortical visual area) with "perception", does not operationally define what "post-perceptual" is, and it ignores a very large existing literature. The results in this manuscript are, in fact, important and interesting, but the conclusions and terms used need to be specific and more carefully chosen. Put another way, even if the fMRI results had been flipped and supported V1 as the sole source of serial dependence, that would not support a "perceptual account" because none of the experiments here provide a new or critical test of "perception" versus memory, decision, etc. The experiments identify a plausible model and neural mechanism in the readout of information from visual cortex. This could hold for serial dependence in perception and/or decision and/or memory. Best to remain agnostic about that.

Review References

Alais, D., Xu, Y., Wardle, S. G., & Taubert, J. (2021). A shared mechanism for facial expression in human faces and face pareidolia. Proceedings of the Royal Society B, 288(1954), 20210966.

Bae, G. Y. (2021). Neural evidence for categorical biases in location and orientation representations in a working memory task. NeuroImage, 240, 118366.

Bliss DP, D'Esposito M (2017) Synaptic augmentation in a cortical circuit model reproduces serial dependence in visual working memory Bazhenov M, ed. PLOS ONE 12:e0188927. 

Cicchini, G. M., Mikellidou, K., & Burr, D. (2017). Serial dependencies act directly on perception. Journal of vision, 17(14), 6-6.

Cicchini, G. M., Benedetto, A., & Burr, D. C. (2021). Perceptual history propagates down to early levels of sensory analysis. Current Biology, 31(6), 1245-1250.

Collins, T. (2019). The perceptual continuity field is retinotopic. Scientific reports, 9(1), 1-6.

Collins, T. (2020). Serial dependence alters perceived object appearance. Journal of Vision, 20(13), 9-9.

Collins, T. (2021). Serial dependence occurs at the level of both features and integrated object representations. Journal of Experimental Psychology: General.

Fornaciai, M., & Park, J. (2018). Serial dependence in numerosity perception. Journal of vision, 18(9), 15-15.

Fischer J, Whitney D (2014) Serial dependence in visual perception. Nat Neurosci 17:738-743 

Fornaciai, M., & Park, J. (2019). Serial dependence generalizes across different stimulus formats, but not different sensory modalities. Vision research, 160, 108-115.

Fornaciai, M., & Park, J. (2020). Neural dynamics of serial dependence in numerosity perception. Journal of Cognitive Neuroscience, 32(1), 141-154.

Fornaciai, M., & Park, J. (2018). Attractive serial dependence in the absence of an explicit task. Psychological Science, 29(3), 437-446.

Fritsche, M., & de Lange, F. P. (2019). The role of feature-based attention in visual serial dependence. Journal of vision, 19(13), 21-21.

Fritsche M, Mostert P, de Lange FP (2017) Opposite Effects of Recent History on Perception and Decision. Curr Biol 27:590-595. 

Gekas, N., McDermott, K. C., & Mamassian, P. (2019). Disambiguating serial effects of multiple timescales. Journal of vision, 19(6), 24-24.

Ho, H. T., Burr, D. C., Alais, D., & Morrone, M. C. (2019). Auditory perceptual history is propagated through alpha oscillations. Current Biology, 29(24), 4208-4217.

Kim, S., Burr, D., Cicchini, G. M., & Alais, D. (2020). Serial dependence in perception requires conscious awareness. Current Biology, 30(6), R257-R258.

Kim, S., & Alais, D. (2021). Individual differences in serial dependence manifest when sensory uncertainty is high. Vision Research, 188, 274-282.

Kiyonaga, A., Scimeca, J. M., Bliss, D. P., & Whitney, D. (2017). Serial dependence across perception, attention, and memory. Trends in Cognitive Sciences, 21(7), 493-497.

Liberman, A., Fischer, J., & Whitney, D. (2014). Serial dependence in the perception of faces. Current biology, 24(21), 2569-2574.

Manassi, M., Ghirardo, C., Canas-Bajo, T., Ren, Z., Prinzmetal, W., & Whitney, D. (2021). Serial dependence in the perceptual judgments of radiologists. Cognitive research: principles and implications, 6(1), 1-13.

Manassi, M., Kristjánsson, Á. & Whitney, D. Serial dependence in a simulated clinical visual search task. Sci Rep 9, 19937 (2019). https://doi.org/10.1038/s41598-019-56315-z

Manassi, M., Liberman, A., Chaney, W., & Whitney, D. (2017). The perceived stability of scenes: serial dependence in ensemble representations. Scientific reports, 7(1), 1-9.

Murai, Y., & Whitney, D. (2021). Serial dependence revealed in history-dependent perceptual templates. Current Biology.

Taubert, J., Van der Burg, E., & Alais, D. (2016). Love at second sight: Sequential dependence of facial attractiveness in an on-line dating paradigm. Scientific reports, 6(1), 1-5.

Taubert, J., & Alais, D. (2016). Serial dependence in face attractiveness judgements tolerates rotations around the yaw axis but not the roll axis. Visual Cognition, 24(2), 103-114.

Papadimitriou C, Ferdoash A, Snyder LH (2015) Ghosts in the machine: memory interference from the previous trial. J Neurophysiol 113:567-577. 

Rafiei, M., Hansmann-Roth, S., et al. Optimizing perception: Attended and ignored stimuli create opposing perceptual biases. Atten Percept Psychophys 83, 1230-1239 (2021). https://doi.org/10.3758/s13414-020-02030-1

Taubert, J., Alais, D., & Burr, D. (2016). Different coding strategies for the perception of stable and changeable facial attributes. Scientific reports, 6(1), 1-7.

Turbett, K., Palermo, R., Bell, J., Burton, J., & Jeffery, L. (2019). Individual differences in serial dependence of facial identity are associated with face recognition abilities. Scientific reports, 9(1), 1-12.

Van der Burg, E., Rhodes, G., & Alais, D. (2019). Positive sequential dependency for face attractiveness perception. Journal of vision, 19(12), 6-6.

Van der Burg, E., Toet, A., Brouwer, A. M., & van Erp, J. B. (2021). Sequential Effects in Odor Perception. Chemosensory Perception, 1-7.

Xia, Y., Leib, A. Y., & Whitney, D. (2016). Serial dependence in the perception of attractiveness. Journal of vision, 16(15), 28-28.

Reviewer #3: Sensory readout accounts for adaptation

Sheehan & Serences

This manuscript uses psychophysics, fMRI, two-photon imaging, and computational modeling to better understand the serial dependence effect in visual perception. Behaviorally, they find a healthy serial dependence effect, wherein the perceived orientation of a stimulus is 'drawn to' the previously presented stimulus orientation in a sequence. However, using both fMRI and analyses of two-photon imaging data in mice, they find evidence in early visual cortex that does not exhibit this attractive effect, instead observing instances in which the orientations are repulsive, a pattern more consistent with an adaptation-based account for visual encoding. The authors interpret these results as evidence against the hypothesis that the perceptual serial dependence effect is a direct result of computations within visual cortex, but rather the result of a readout mechanism that is potentially 'aware' of the adapted state of the visual system. The breadth of the methods used to address this phenomenon is impressive, to say the least, and the rigor is generally very high. I have a few questions and concerns that could improve the clarity of the manuscript, which are detailed below.

1. As the authors know, the sluggish nature of the hemodynamic response is problematic when making claims regarding recent past events influencing measures of present events, and how it was potentially tainting the measure of present events with preceding events —an effect that complicates measures of history-dependent serial orientation effects. Different orientations produce different, sometimes significant, changes in BOLD response, and as the authors note, this is a potential serious concern, and they do their best to assuage this concern. I find as a whole this is satisfactory, particularly taking into account the two-photon CA imaging results. However, there is one analysis that was not entirely clear, and I think the authors could better unpack. The authors deconvolve their data, and then fit these deconvolve responses with a double gamma, from which they then use the weights for subsequent analyses. But, one thing I need clarification on is whether this was a deconvolution that was stimulus orientation dependent, or across all orientations. I ask because I want to better understand whether they observed any qualitative changes between various orientation conditions at all. Some more clarification of this analysis would help me better understand how appropriately it addresses the issue.

2. The analysis of the Stringer et al (2019) paper very nicely supports the conclusions of the authors. One detail (or analysis) that would help bolster this would be to ensure the reader that the statistics of the sequencing of stimulus events themselves were such that the repulsive biases were truly the result of neural processes, and not an accidental result of orientation sequences that have some non-randomness to them that could have given rise to the observed effects. I imagine it's not a huge concern, but some more detail about those stimuli and the sequencing would be important here. 

3. The model that the authors put forth is interesting, but some aspects of it are puzzling and could use a bit more substantiating or caveats. In particular, I found myself puzzling over what a 'bases aware or hyper-aware' schema would look like, and how that would be possibly implemented. In particular, were these plausible scenarios? It would seem the "aware" class of models involve the system possessing a population that maintains a *veridical representation*, and can choose to compare it to the affected representation? I acknowledge this is just a model, but it seemed to me a bit strange for there to be a scenario in which this were implemented in the brain. I think the authors could do a better job substantiating these hypothetical alternative models, backing the possibilities up a little with more empirical precedent.

4. The 'awareness' component of this study, in particular its modeling component, should reference a recent paper by Kim, Burr, Cicchini & Alais (2020) that found that serial dependence did not emerge when stimuli were suppressed under binocular rivalry. This supports the notion put forth by the authors that this effect transpires later in the perceptual processing stream.

---

## [Decision Letter · Decision Letter 2]

4 May 2022

Dear Dr Sheehan,

Thank you for submitting your revised Research Article entitled "Sensory readout accounts for adaptation" for publication in PLOS Biology. I have now obtained advice from the original reviewers and have discussed their comments with the Academic Editor. 

Based on the reviews, we will will move towards acceptance of this manuscript for publication, provided you satisfactorily address the remaining points raised by Reviewer 1. We also ask that you consider some minor modifications to the title and abstract to increase the accessibility of this work for our broader readership and, when going through the revision for this final submission, that you ensure the writing allows for broad access to this interesting work. Please also make sure to address critical data and other policy-related requests that are required before we can proceed with acceptance (see bottom of this email for these points). 

Title: We'd like you to consider a title change that makes it clear that this work is addressing neural correlates of perceptual adaptation. For example, something along the lines of: "Neural adaptation during sensory encoding gives rise to perceptual bias"

Abstract: We'd also suggest the modifications to the abstract with something along the following lines:

Sensory behaviors and neural responses to stimuli are strongly shaped by prior stimulus exposure. In the case of visual perception, an individual’s report of how they perceive a visual stimulus can sometimes show a bias towards previously viewed stimuli (i.e. serial dependence). Previous behavioral studies suggest that serial dependence is implemented via modulations in visual cortex, but neural evidence is lacking. To address this, we recorded fMRI responses while human participants (male and female) performed a delayed orientation discrimination task and utilized 2-photon imaging of mouse visual cortex in animals engaged in a related discrimination task. While the behavioral reports of participants demonstrate an attractive bias towards previous stimuli, neural response patterns in sensory areas exhibit a repulsive bias. A similar repulsive neural bias was found in mouse visual cortex. Using a model where both sensory encoding and readout are shaped by stimulus history, we show that neural adaptation at the encoding phase reduces redundancy and leads to the repulsive neural biases that we observed in visual cortex. Thus, our model suggests that serial dependence is not implemented in visual cortex but rather that the visual system improves neural efficiency via readout mechanisms that account for adaptation during encoding while still optimizing behavioral readouts based on the temporal structure of natural stimuli. 

As you address these items, please also take this last chance to review your reference list to ensure that it is complete and correct. If you have cited papers that have been retracted, please include the rationale for doing so in the manuscript text, or remove these references and replace them with relevant current references. Any changes to the reference list should be mentioned in the cover letter that accompanies your revised manuscript.

We expect to receive your revised manuscript within two weeks. 

*Published Peer Review History*

*Press*

Sincerely,

Kris

Kris Dickson, Ph.D.

Neurosciences Senior Editor/Section Manager,

kdickson@plos.org,

PLOS Biology

ETHICS STATEMENT: Fine as is

DATA POLICY: The raw data underlying all individual quantitative observations summarized in the main and supplemental figures and results of your paper must be made available at this stage. 

You may be aware of the PLOS Data Policy, which requires that all data be made available without restriction: http://journals.plos.org/plosbiology/s/data-availability. For more information, please also see this editorial: http://dx.doi.org/10.1371/journal.pbio.1001797. 

2) Deposition in a publicly available repository. If you choose to deposit this information in a repository like Zenodo, please generate the DOI now even if you do not release it until publication. We require an accession code or a reviewer link at this stage so that we may view your raw data before publication. 

Regardless of the method selected above, please ensure that you provide the individual numerical values that underlie the summary data displayed in the following figure panels as they are essential for readers to assess your analysis and to reproduce it:

Fig 1B-H; Fig2 all (A-I), Fig3 all (A-F), Fig 5 all (A-G)

Supplemental Figures: 1A-D, 2 all (A-J), 3 all (A-F), Fig 4 all (A-E), Fig 5 all (A-D), Fig 6 all (A-D), Fig 7 all (A-D), Fig 8 all

IMPORTANT (and often forgotten!): Please also ensure that FIGURE LEGENDS IN YOUR MANUSCRIPT AND SUPPLEMENTAL FILES must include statements directing readers on where the underlying data can be found. Please also ensure your supplemental data file/s has a legend.

CODE AVAILABILITY

Upon publication of your article, we ask that you share publicly any code that you created and that directly relates to the results described in your article, unless you claim an exemption to the policy.

If you have legal or ethical restrictions on public sharing of your code, please include details of your exemption in your Data Availability Statemen for us to consider.

A statement about where and how your code can be accessed must be included in the Data Availability Statement in your manuscript.

DATA NOT SHOWN?

Reviewer remarks:

Reviewer's Responses to Questions

PLOS authors have the option to publish the peer review history of their article (what does this mean?). If published, this will include your full peer review and any attached files.

Reviewer #1: No

Reviewer #2: No

Reviewer #3: No

Reviewer #1: The current version of the manuscript is substantially better streamlined and clarified. A major difference is that a previous result in rats was found to be spurious, but that does not harm the otherwise solid results in humans - perhaps it even improved the readability of the paper, since that result was tangential, anyway. 

I have a few minor comments, but I fully support the publication of the study. Congratulations on this very interesting study.

1. "The identity of the previous stimulus for spatial position and angle has been shown to be decodable from the spiking activity of single units in the frontal eye field (FEF)" 

Here PPC in rats (Akrami at al) and PFC in monkeys (Barbosa et al) are other instances of previous stimulus decodability from spiking activity. 

2. "It is curious that unlike some spatial working memory studies (Papadimitriou et al., 2015; Bliss et al., 2017;

Stein et al., 2020), we did not find that behavioral biases increased with delay time. One possible

explanation is that this phenomenon is actually unique to spatial working memory due to either a more

consistent increase in sensory uncertainty of spatial location due to eye movements or a separate

mechanism of memory maintenance that becomes more susceptible to proactive interference relative to

orientation memories."

I appreciate that the authors now fully acknowledge the discrepancy, but I don't think that different experimental modalities explain the differences. We have seen this effect in other modalities (unpublished), but see Fritsche (2017) for a very similar task. Finally, there is even recent theoretical arguments for this effect: https://www.biorxiv.org/content/biorxiv/early/2022/03/02/2022.02.28.482269.full.pdf

3. I find it a bit surprising that the new finding "decoding using sensory localizer data was consistently at chance during N+1 trial suggesting information relating to past stimulus is not stored in a sensory code." did not make it to the main manuscript. This is directly related with the title of the manuscript and claims raised in the abstract, so I think it should be giving its deserved attention in the main manuscript.

4. About my initial comment on 183 that you said it would be circular: I might be missing something, but I don't see why. 

What you do now is plotting (prev_stimulus - curr_stimulus) vs (pred_curr_stimulus - curr_stimulus). 

What I was curious about was (pred_curr_stimulus - curr_stimulus) vs (pred_prev_stimulus - curr_stimulus) or vs (pred_prev_stimulus - pred_curr_stimulus). 

pred = predicted from neural activity

Do you think this would be circular?

In any case, this is mostly my own curiosity, rather a requirement for acceptance.

Reviewer #2: The authors should be commended for the excellent and thorough revisions. All of the comments were addressed and the manuscript is much stronger, clearer, and more balanced. I recommend acceptance without delay.

Reviewer #3: The authors have done a great job addressing my concerns, and I recommend this for publication

signing my review, Sam Ling

---

## [Editor Report · Decision Letter 3]

14 Jun 2022

Dear Tim,

Thank you for the submission of your revised Research Article "Attractive serial dependence overcomes repulsive neuronal adaptation" for publication in PLOS Biology. On behalf of my colleagues and the Academic Editor, Frank Tong, I am pleased to say that we can in principle accept your manuscript for publication, provided you address any remaining formatting and reporting issues. 

Most of these will be detailed in an email you should receive within 2-3 business days from our colleagues in the journal operations team. Editorially, there is one minor formatting issue that also needs to be addressed - while you have nicely directed readers to the OSF file in the figure legends in the main body of your manuscript, we also need you to add this statement to each of the relevant Supplemental Figure legends. Please note that we will not be able to formally accept your manuscript and schedule it for publication until you have completed this and any requested changes from the operations team.

PRESS

We frequently collaborate with press offices. If your institution or institutions have a press office, please notify them about your upcoming paper at this point, to enable them to help maximize its impact. If the press office is planning to promote your findings, we would be grateful if they could coordinate with biologypress@plos.org. If you have previously opted in to the early version process, we ask that you notify us immediately of any press plans so that we may opt out on your behalf.

Sincerely, 

Kris

P.S. I apologize for how long it took me to get this decision letter out to you. Frank had wanted to have one last look over the revision and then I was traveling to a meeting.

Kris Dickson, Ph.D. (she/her)

Neurosciences Senior Editor/Section Manager

PLOS Biology

kdickson@plos.org